# Microfluidics combined with electron microscopy for rapid and high-throughput mapping of antibody–viral glycoprotein complexes

Leigh M. Sewall[1], Rebeca de Paiva Froes Rocha[1,2], Grace Gibson[1], Michelle Louie[1], Zhenfei Xie[3], Sandhya Bangaru[1], Andy S. Tran[1], Gabriel Ozorowski[1], Subhasis Mohanty[4], Nathan Beutler[5], Thomas F. Rogers[5,6], Dennis R. Burton[5], Albert C. Shaw[4], Facundo D. Batista[3,7], Blanca Chocarro Ruiz[1], Alba Torrents de la Peña[1] & Andrew B. Ward[1] ✉

Understanding the mechanistic interplay between antibodies and invading pathogens is essential for vaccine development. Current methods are labour and time intensive and limited by sample preparation bottlenecks. Here we present microfluidic electron microscopy-based polyclonal epitope mapping (mEM), which combines microfluidics with single-particle electron microscopy for the structural characterization of immune complexes using small volumes of sera (<4 µl). First, we used mEM to map polyclonal antibodies present in sera from infected and vaccinated individuals against five viral glycoproteins using negative-stain electron microscopy. The mEM detected a greater number of epitopes compared with conventional polyclonal epitope structural mapping methods. Second, we used mEM and cryo-electron microscopy to characterize two coronavirus spikes and one HA glycoprotein with and without polyclonal antibodies. Finally, we mapped individual antibody responses over time in mice vaccinated with human immunodeficiency virus envelope N332-GT5. mEM enables the rapid, high-throughput mapping of antibodies targeting a broad range of glycoproteins, facilitating a better understanding of infection and guiding structure-based vaccine design.

Identifying the molecular interactions between antibodies and viral pathogens provides important insight into possible mechanisms of protection during infection or vaccination. The high-throughput determination of potential sites of vulnerability on viral glycoproteins such as human beta coronavirus (CoV) spike, influenza haemagglutinin (HA) and human immunodeficiency virus envelope (HIV Env) can therefore enable rapid structure-guided vaccine (re)design and confer vital understanding of the progression of a response to infection in real time[1]. Conventional serological techniques are typically used to characterize these interactions. Enzyme-linked immunosorbent assays (ELISAs) measure total antibody binding to viral glycoproteins[2] but do not reveal the epitopes targeted without the laborious creation of epitope knockout mutants that require a priori knowledge and might not comprehensively define all responses. Neutralization assays can measure the inhibition potential of serum antibodies[3] but miss antibodies that are not functionally relevant. The isolation of monoclonal antibodies through B cell sorting can provide antibody sequence as well as structural insight into sites of vulnerability[4]; however, solving

structures of individual complexes is time consuming and often underestimates the epitopes targeted.

More recently, electron-microscopy-based polyclonal epitope mapping (EMPEM) paved the way for the broad evaluation of polyclonal immune responses to viral glycoproteins[5–7]. Using this technique, polyclonal antibodies are isolated from sera, digested into fragment antigen binding (Fab) regions and complexed with glycoproteins, which are then visualized by negative-stain electron microscopy (ns-EM) or cryo-electron microscopy (cryo-EM)[8]. However, sample purification is extremely time consuming and laborious, requiring over a week for a single serum sample, as well as specialized personnel. Further, EMPEM necessitates large sample volumes (>0.5 ml), precluding those available from biobanks[9] and important preclinical animal models[10,11], and confining large-scale clinical studies to fewer patients.

As EMPEM becomes integrated into vaccine evaluation workflows, scalability, throughput and miniaturization are important for wider adoption of the technique. The field of microfluidics therefore offers a promising solution for decreased sample and reagent consumption, reduced hands-on time, higher throughput capabilities and automatization[12–15]. Recent success miniaturizing cryo-EM sample preparation using microfluidics for characterizing time-resolved binding intermediates[16], improving protein integrity[17] and purifying protein directly on functionalized grids[18] lends further support for the development of a less laborious and more efficient system for in-depth structural elucidation of protein–protein interactions.

Here, we describe microfluidic electron microscopy (EM)-based polyclonal epitope mapping (mEM), a rapid, high-throughput technology for the structural characterization of polyclonal immune responses to infection and vaccination. We demonstrate that mEM can capture six different viral glycoproteins for characterization by ns-EM and cryo-EM. Using minimal volumes of sera (<4 µl), we show that mEM enables faster sample preparation (90 min) for a high-throughput assessment of both memory- and vaccine-elicited antibody responses to a panel of viral glycoproteins, compared with conventional EMPEM methodology. Further, in using IgG rather than Fab, mEM increases the detection of polyclonal antibody responses. Finally, we applied this technology to an HIV vaccine trial in mice and show the longitudinal development of polyclonal antibody responses at the individual level, crucial for the redesign of vaccine candidates in preclinical studies.

## Results

### Development of mEM technology

The mEM technology uses a microfluidic platform for the reversible immobilization of viral glycoproteins, enabling their structural and biochemical characterization using single-particle EM (Methods and Fig. 1a). Rapid structural evaluation of these immune complexes requires minimal sample volumes and reagents as well as a streamlined approach to their capture. To achieve this, we first designed small-volume flow cells using polydimethylsiloxane (PDMS) polymer[14] that integrates with a biocompatible gold surface containing scaffolds of self-assembled monolayers (SAMs)[19] (Fig. 1b). These SAMs are composed of 16-mercaptohexadecanoic acid (MHDA) and are covalently linked to a capture protein Strep-TactinXT, the engineered variant of streptavidin (Methods and Supplementary Fig. 1).

Next, to enable the capture of immune complexes, we produced recombinantly expressed glycoproteins containing a Twin-Strep-tag, which binds the immobilized Strep-TactinXT with picomolar affinity[20]. The blocking agent bovine serum albumin (BSA) is included before the injection of patient sera (4 µl, <1% of material required for conventional EMPEM) to avoid non-specific adsorption. Once the polyclonal antibodies present in sera bind the targeted viral glycoproteins, the immune complexes are eluted through competition with biotin[21] and structurally characterized by ns-EM or cryo-EM (Methods, Fig. 1d and Supplementary Fig. 2). Using mEM, the preparation of patient immune complexes, before imaging and data analysis, requires approximately

90 min, a substantial improvement from the conventional sample preparation methodology requiring at least 1 week. Overall, the whole process, from sample preparation to imaging and data analysis, can take an additional 4–24 h, representing a total time of 1–1.5 days.

To optimize the MHDA-linked surface conditions, we expressed spike glycoprotein (S) from the pandemic strain severe acute respiratory syndrome coronavirus 2 (SARS-CoV-2) and first used surface plasmon resonance (SPR), a standard tool for the study of biomolecular interactions[22]. Our results demonstrated that a concentration of 2.5 mM MHDA was sufficient to form dense SAM scaffolding on the gold surface with immobilization of Strep-TactinXT at 100 µg ml$^{-1}$ yielding the largest signal response of 2,059 resonance units (RU) (Supplementary Fig. 3a). In addition, S injected at 1 mg ml$^{-1}$ resulted in maximum binding of glycoprotein (Methods and Supplementary Fig. 3b). These conditions were then extrapolated to mEM where we used ns-EM to profile the capture efficacy of the surface. First, we observed an increase in S recovery with a higher concentration of 5 mM MHDA (Supplementary Fig. 4a,b). Particle density of the eluted fraction ranged from 75 to 100 S particles per micrograph (Fig. 1e), which corresponds to a conventional ns-EM image with particles diluted to 20 µg ml$^{-1}$, or an average of 90–125 particles per micrograph (Fig. 1f). This density is generally suitable for efficient data collection and downstream three-dimensional (3D) reconstruction. The biofunctionalized gold could be recycled more than four times and the PDMS cell can be reused up to 1 year (Methods), making the mEM device entirely reusable.

### Capture and characterization of viral glycoproteins

To examine mEM capture efficiency for low- and high-resolution glycoprotein characterization, we expressed an expanded panel of surface glycoproteins from seasonal and pandemic viral strains (Methods and Fig. 2a). Among the CoV strains, we selected seasonal OC43 and HKU1 variants of S, as well as SARS-2, and used ns-EM to ensure structural integrity and appropriate protein density on the EM grid. For all CoV glycoproteins, the particle density ranged from 75 to 155 particles per micrograph image (Fig. 2b), which is comparable to the previously described standard ns-EM protein micrograph density (Fig. 1g). In addition, seasonal strains of influenza HA, H1 (Michigan/045/15) and influenza B (Maryland/15/16), were similarly captured with mEM; however, a lower particle density was observed (20–45 particles per micrograph) (Fig. 2c). Finally, the capture of engineered HIV Env trimer (named N332-GT5)[23] yielded a particle range of 200–270 particles per micrograph (Fig. 2d), consistent with a standard ns-EM dataset.

Achieving a meaningful resolution using cryo-EM requires specific sample preparation conditions are met, including high protein concentration to ensure sufficient particle numbers per image are collected. As such, we performed single-particle cryo-EM on captured OC43 S, influenza B HA and SARS-2 S glycoproteins to assess cryo-EM compatibility with mEM. A mixture of OC43 and influenza B HA (1:1) was used to determine high-resolution structures of both proteins. The three-fold symmetric (C3) reconstruction of apo OC43 S trimer resulted in a 3.3-Å-resolution map, with all receptor binding domains (RBDs) found in a closed conformation, similar to previous studies[6]. Superimposition of the published atomic models with the model built for this map revealed a C-alpha r.m.s.d. of 0.43 Å (Fig. 2e and Supplementary Fig. 5a). In addition, the C3 reconstruction of influenza B HA generated a 3.0-Å-resolution map (Fig. 2f). SARS-2 S revealed the presence of different classes: S with open or closed RBD conformation. The C3 reconstruction of S with a closed conformation resulted in a 4.6-Å-resolution map (Fig. 2g), whereas the asymmetric reconstruction (C1) of S in a partially open conformation reconstructed to 6.9 Å resolution (Fig. 2h), although both were limited by low particle numbers. These results establish that mEM is compatible with cryo-EM using only small amounts of glycoproteins, although high resolution benefits from lower heterogeneity within the dataset.

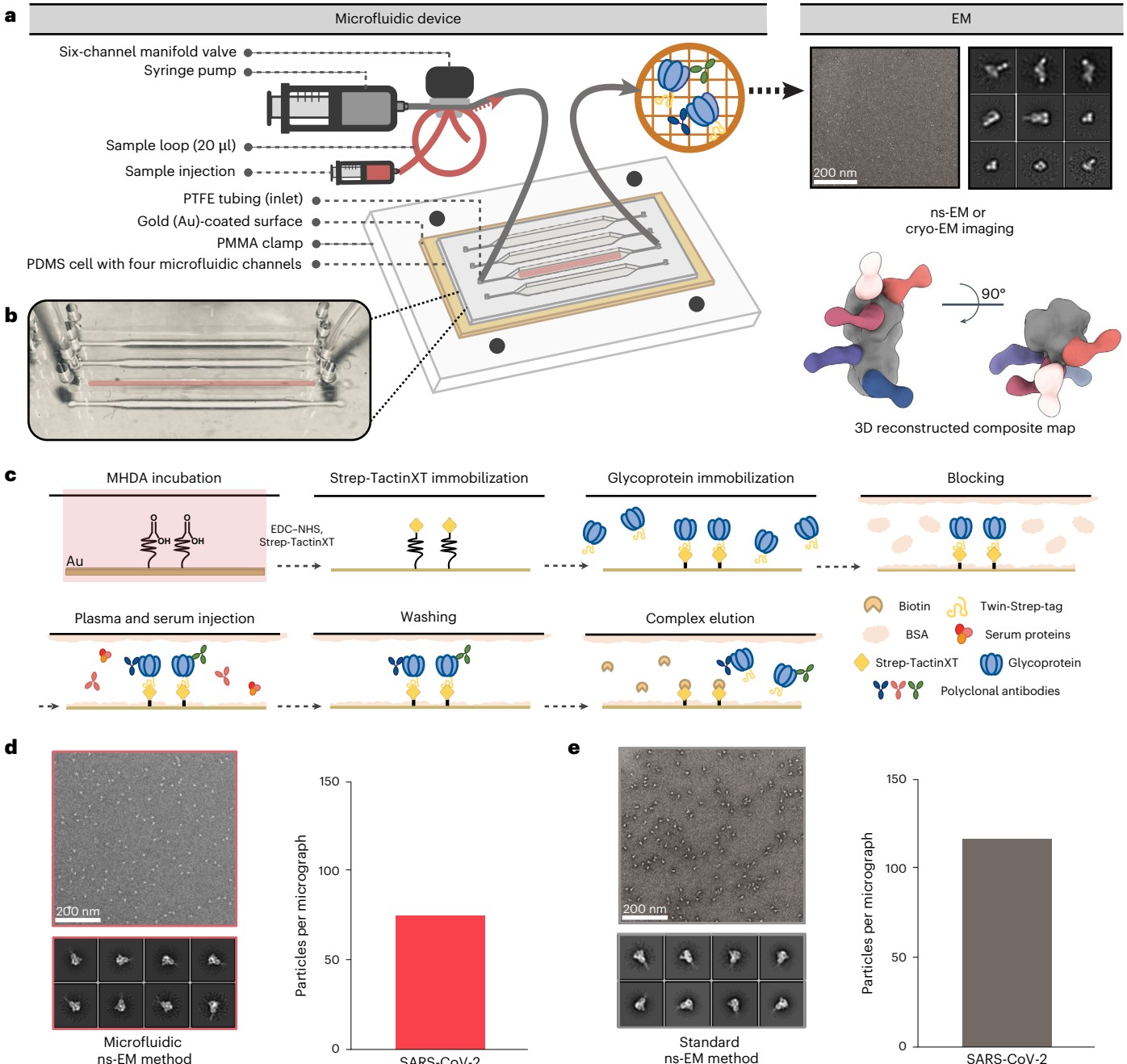

**Fig. 1 | mEM is a platform that combines microfluidics and EM to characterize immune responses to infection or vaccination. a**, Left: a schematic of microfluidic technology demonstrating running buffer injected by the syringe pump and passed through a channel (red). Additional components include the input port and loop for sample injection of glycoproteins and sera, manifold valve, gold surface and PDMS cell. mEM outputs glycoprotein–antibody complexes that are imaged using EM for epitope characterization. Right: an exemplar micrograph, 2D class averages and 3D reconstructions of influenza HA complexed with IgG from sera. **b**, A photograph of the experimental PDMS cell with sample flowing through one of the channels (red). **c**, A representation of the seven main steps used by the mEM technology in preparing immune complexes for structural characterization by EM. **d**, The immobilized SARS-CoV-2 spike glycoprotein is eluted, and particles per micrograph are assessed by ns-EM. **e**, The particle density of the eluted sample is compared with a standard concentration of 15 μg ml$^{-1}$ used to perform ns-EM. Representative graphs for one spike glycoprotein experiment out of three replicates ($n$ = 3 experiments) are shown for **d** and **e**.

## Structural analysis of antibody–glycoprotein binding

To demonstrate that mEM could be used to purify immune complexes for structural characterization, we used monoclonal and purified polyclonal antibodies that target the SARS-CoV-2 S. First, we examined two previously published monoclonal antibodies: TXG-0078, which binds the N-terminal domain (NTD) supersite region[24], and CC6.30.2, which binds the RBD site B (class 2)[25]. To assess mEM compatibility with both IgG and Fab, we selected TXG-0078 Fab and CC6.30.2 IgG. Using mEM to capture the antibody–S complexes and ns-EM to visualize the complexes, we observed that TXG-0078 was bound to the NTD supersite on all three protomers and CC6.30.2 was bound to all RBDs on S (Fig. 3a and Supplementary Fig. 6a,b), similar to the high-resolution structures previously published. These results indicated that antibody–glycoprotein complexes could be readily captured by mEM and eluted for structural characterization.

Prior efforts to structurally determine immune complexes with polyclonal IgG have been hindered by the propensity of IgG to cross-link multiple glycoproteins and induce large protein aggregates

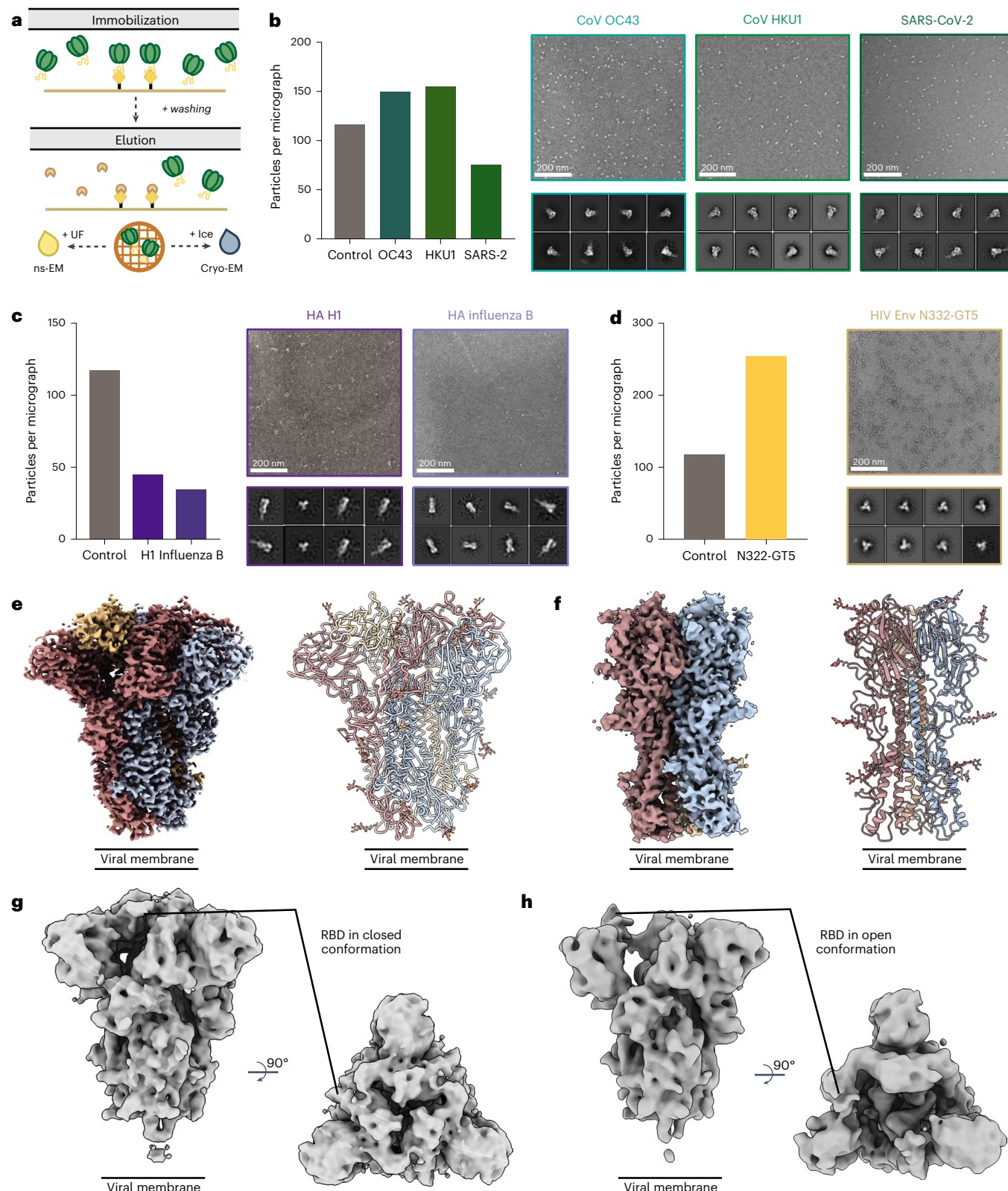

**Fig. 2 | Immobilization of a broad array of viral glycoproteins using mEM.**
**a**, A schematic of SARS-CoV-2 S immobilization and elution, which is directly added to copper mesh grids for ns-EM analysis or graphene oxide grids for cryo-EM analysis. **b**–**d**, The number of particles per micrograph in ns-EM, directly from the elution fractions of CoV spike (**b**), influenza HA (**c**) and HIV Env N332-GT5 (**d**). Experiments were performed with mEM using 35 µg ml⁻¹ of purified protein and analysed by ns-EM. Exemplar micrographs and 2D class averages are shown for each glycoprotein. For all glycoprotein immobilizations, representative graphs of a single experiment out of three replicates (*n* = 3 experiments) are shown. **e**, A side view of the cryo-EM maps and atomic models, represented as a ribbon, of CoV OC43 spike in the prefusion state (**e**) and HA influenza B/Maryland/15/16 (**f**). The protomers of both glycoproteins are coloured in red, blue and yellow for clarity. **g,h**, Side and top views of the cryo-EM maps of SARS-CoV-2 showing the S in prefusion state with the RBDs in the closed ('down') conformation (**g**), and with one RBD in the open ('up') conformation (**h**). Densities of both maps are shown in grey.

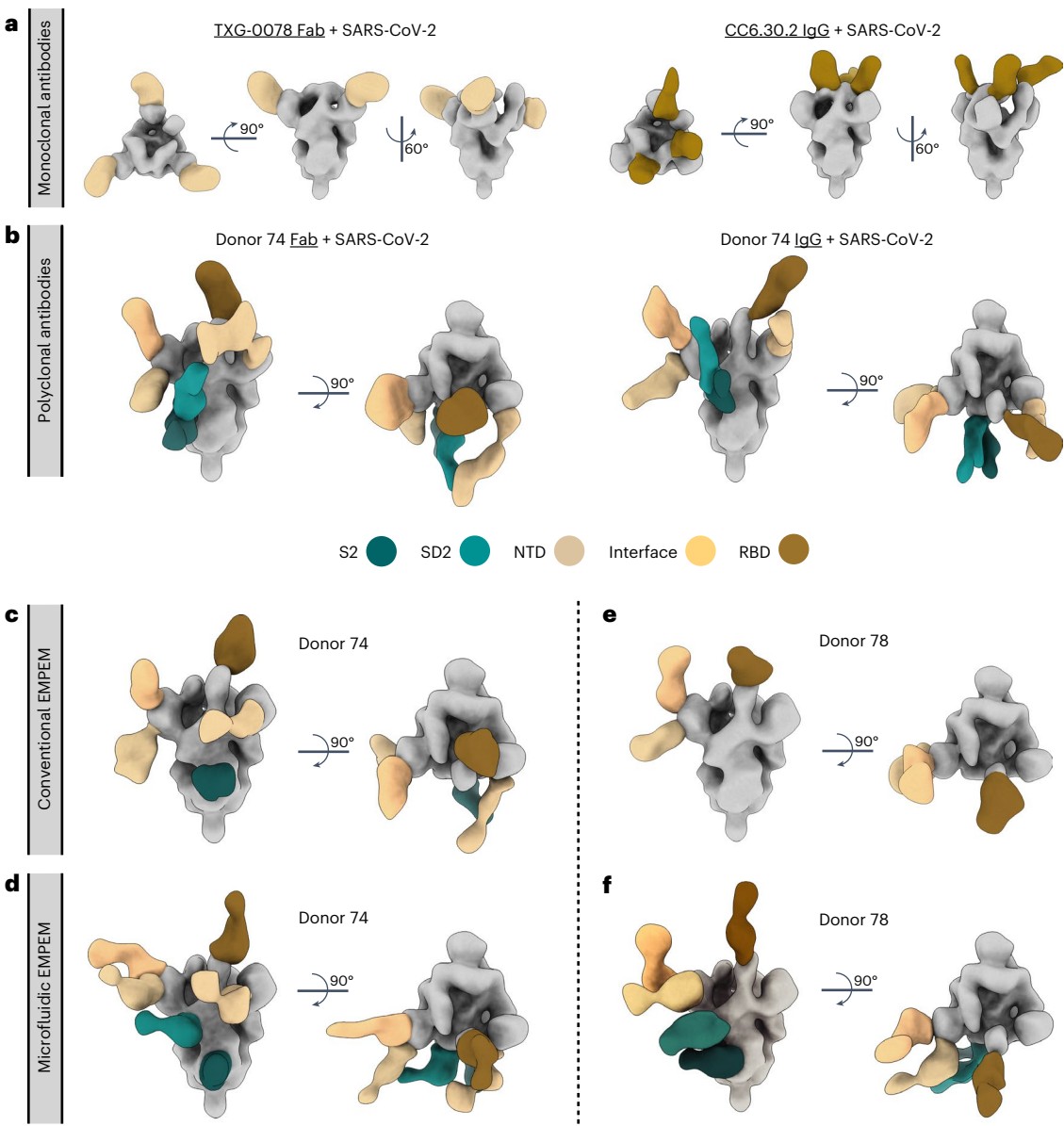

**Fig. 3 | mEM polyclonal epitope mapping using purified antibodies and human sera from individuals infected with SARS-CoV-2. a**, Monoclonal antibodies in the form of Fab or IgG (underlined) were passed through the channel containing captured SARS-CoV-2 S, and the eluted antibody–glycoprotein complexes were imaged by ns-EM. Side and top views of segmented 3D maps are shown. **b**, Purified polyclonal antibodies (underlined) —in the form of Fab or IgG—were assessed for binding using mEM. Side and top views of segmented composite maps for both immune complexes are shown. **c–f**, Polyclonal epitope mapping using sera from two patients previously infected with SARS-CoV-2 by ns-EM. Side and top views of segmented composite 3D maps generated using conventional EMPEM (**c** and **e**) and mEM (**d** and **f**) methods are shown for both donors.

(Supplementary Fig. 7). Therefore, to test whether mEM provides more favourable conditions to prevent aggregation, namely a fast elution time, we obtained sera from a human donor (donor 74) who was infected with SARS-CoV-2 before vaccination and had potent neutralizing antibody titres against the virus (inhibitory dilution with 50% virus neutralization, $ID_{50}$ 7791). Polyclonal antibodies were isolated from the sera before loading and complexed with captured S for epitope mapping using mEM. By ns-EM, both polyclonal IgG and Fab were characterized targeting identical sites, including S2, subdomain 2 (SD2), RBD and three NTD sites (Fig. 3b and Supplementary Fig. 6c,d), with no substantial aggregation observed. We confirmed these epitopes using conventional, large-volume EMPEM with purified polyclonal Fab (Methods, Fig. 3c and Supplementary Fig. 6e), observing a similar landscape of mapped sites, with the notable exception of SD2, which was detected only with mEM using polyclonal IgG and Fab.

Importantly, these findings demonstrated that mEM could enable the capture of isolated polyclonal IgG without inducing glycoprotein–antibody aggregation.

After establishing the use of mEM for polyclonal epitope mapping, we sought to detect immune responses directly from sera. Conventional EMPEM typically requires purification of antibodies before immune complex formation[8]. Here, we hypothesized that the same molar excess of polyclonal antibody to glycoprotein from a small volume of human sera (~4 μl) could provide similar complexing conditions by mEM (Methods). To expand our analysis, we acquired sera from a second human donor (donor 78) infected with SARS-CoV-2 before vaccination with similar neutralizing ability ($ID_{50}$ 2956) and first performed epitope mapping using conventional EMPEM with purified polyclonal Fab. Using ns-EM, we identified antibodies that target the NTD and RBD regions on S (Fig. 3d and Supplementary Fig. 6f). Next, using

mEM, we injected 4 µl of sera from both donors over captured S and characterized the immune complexes by ns-EM (Methods and Fig. 1c). Here again we observed additional epitopes, SD2 and S2 (Fig. 3e,f and Supplementary Fig. 6g,h) relative to conventional EMPEM. To confirm mEM reproducibility, we followed the same methodology with sera from donor 74 in triplicate and observed a similar epitopic landscape across all three complexes, including the additional SD2-targeting antibody (Supplementary Fig. 8). Models of SARS-2 S (PDB 6VYB) and a human polyclonal Fab with a polyalanine backbone were docked into the final map, and we observed the models fit well within each of the Fab densities (Supplementary Fig. 13a). Taken together, these findings demonstrate the use of polyclonal IgG directly from sera, and the improved sensitivity with mEM results in a rapid and material-efficient technology for polyclonal antibody detection.

### Rapid analysis of memory- and vaccine-elicited responses

To determine whether mEM could be used for high-throughput epitope mapping, we assessed the sample preparation capability of mEM in comparison with conventional EMPEM methodology. First, we expressed a panel of viral glycoproteins to assess both preexisting and acute vaccine-elicited antibody responses. Four seasonal viral strains were selected, including the CoV S strains OC43 and HKU1 and HA strains A/Michigan/045/2015 and B/Maryland/15/16, all previously characterized with mEM. Next, we obtained sera from four individuals (donor 1: 2323, donor 2: 2333, donor 3: 2327, donor 4: 2336) enrolled in an influenza vaccination study. All donors received a high-dose trivalent influenza vaccine (Fluzone) containing the two expressed HA strains and administered at day 0, with sera collected at days 0, 2, 7, 28 and 70 (Supplementary Fig. 9a). Corresponding haemagglutination inhibition (HAI) and ELISA data confirmed the presence of influenza-specific antibodies (Supplementary Fig. 9b–d), while preexisting responses to the seasonal CoVs remained unknown. We selected one timepoint per individual (day 0: 2333 and 2336, day 2: 2327, day 7: 2323) where enough sera were available to perform mEM and conventional EMPEM.

We first examined the presence of preexisting antibodies from prior seasonal CoV infection, a potential indicator of SARS-CoV-2 infection outcome[26–28]. For all four donors, we observed a broad range of epitopes targeted on both OC43 and HKU1 S using mEM, indicating that all donors had preexisting antibody responses to both seasonal CoVs before the coronavirus disease 2019 (COVID-19) pandemic. These results were confirmed using conventional EMPEM; however, the level of epitope detection was notably higher by mEM with additional epitopes detected in all four donors against both CoVs (Fig. 4a and Supplementary Fig. 10). In addition, rigid-body fits docking in models of OC43 S (PDB 6OHW), HKU1 S (PDB 5I08) and a human polyclonal Fab with a polyalanine backbone fit well within each of the Fab densities (Supplementary Fig. 13b,c). Epitope mapping of donor 2327 illustrates the improved sensitivity of mEM: using mEM, polyclonal antibodies present in sera were detected against S2, NTD and the CTD of OC43, while only S2- and CTD-directed antibodies were detected with conventional EMPEM. In addition, although no antibodies were found to target HKU1 by conventional methodology, the S2 and CTD sites were detected with mEM (Fig. 4b, left). We next used mEM in combination with cryo-EM to perform high-resolution structural mapping of the donor 2327 immune complex with OC43 S and donor 2323 immune complex with influenza B HA. Although the particle density and the abundance of antibodies per epitope were low, this approach allowed us to resolve low-resolution structures of a CTD antibody in OC43 and two distinct antibodies binding to the low stem HA epitopes, respectively (Fig. 4b, bottom).

Next, to reveal immune responses elicited during the vaccination regimen, we used mEM to examine polyclonal antibodies targeting H1 influenza HA and influenza B HA. Responses were assessed in two donors at day 0 and one donor at days 2 and 7. Epitope mapping of all four donors revealed a range of polyclonal antibodies targeting both H1 and influenza B, including stem, esterase, protomer–protomer interface, side head and receptor binding sites, correlating well with the characterization by conventional EMPEM (Fig. 4c and Supplementary Fig. 11). Models of HA H1 (PDB 7KNA) and a human polyclonal Fab with a polyalanine backbone were docked into the final map, and we observed the models fit well within each of the Fab densities (Supplementary Fig. 14a,b). Notably, the protomer–protomer interface sites have not been characterized in previous EMPEM studies; however, the epitope footprint at low resolution appears similar to HA-targeting antibody S8V1-157 (ref. 29). Again, additional sites were detected with mEM for both HA strains in almost every case. Epitope mapping of donor 2323 highlights this finding with side-head epitopes of H1 and influenza B, respectively, identified only when using mEM (Fig. 4b, right). For all four donors, approximately 0.8% of sera (16 µl versus 2 ml total volume) was used to structurally elucidate both memory- and vaccine-elicited antibody responses to multiple viral glycoproteins in a fraction of the sample preparation time previously required (4 days versus >3 weeks for all 16 complexes).

### Longitudinal polyclonal antibody mapping in individual mice

Structure-guided vaccine (re)design aims to define specific surfaces on viral glycoproteins that can elicit protective antibody responses[30], with the chosen vaccine candidates further evaluated in animal models, such as mice, rabbits and non-human primates[31]. Due to the extremely limited sera volumes available from mice, however, EMPEM has been possible only on a group level via pooled sera from six to eight individuals, limiting our understanding of the kinetics of individual responses. To explore this, we used sera from mice vaccinated with N332-GT5 (ref. 32) (Fig. 2d). Two groups of five BL6 mice (group 1: wild type (WT), group 2: BG18[gH] adoptively transferred) were injected with N332-GT5 mRNA at day 0 and sera were collected at days 12, 28 and 42 (Fig. 5a). We selected three mice from each group with the highest ELISA titres (Fig. 5b), which indicated the presence of N332-GT5-specific antibodies. Due to the propensity of polyclonal IgG to cross-link the HIV Env glycoprotein, all mouse sera were diluted 1:100 instead of 1:10 for complexing (Methods and Supplementary Fig. 12a). Our results from mEM reveal the presence of N332-GT5-specific antibodies in each of the three WT group mice by day 28, specifically those targeting the V1/3 loops and base epitopes, with both sites observed in each individual mouse by day 42 (Fig. 5c and Supplementary Fig. 12b,c and 14c). Similarly, for the group that received BG18[gH] B cells, we observed antibodies targeting the V1/3 epitope in all mice at each timepoint, except for one mouse at day 12 (Fig. 5d and Supplementary Fig. 12d). Rigid-body fits docking in models of MD64 N332-GT5 SOSIP (PDB 8T4K) and a human polyclonal Fab with a polyalanine backbone fit well within each of the displayed Fab densities (Supplementary Fig. 14c). These results corroborate the EMPEM findings from an identical N332-GT2 immunization study in which sera from six to seven mice were pooled for analysis at days 14 and 42 (ref. 23). Despite the minute volumes of sera obtained from a single blood draw, mEM at the individual mouse level can now be used in concert with complementary serological approaches such as ELISA and neutralization to determine epitope-specific correlates in mice.

### Discussion

The version of mEM described herein includes several improvements over conventional EMPEM, including minute sera requirements as well as improved speed, throughput of sample processing and epitope detectability. The ability of mEM to detect polyclonal IgG binding directly in sera at a microlitre scale makes it an efficient and sensitive tool for the low- and high-resolution characterization of antibody responses to viral invasion. Furthermore, the rapid progression of EM imaging software[33], grid preparation[34–36] and incorporation of machine learning into data processing pipelines[37,38] can be seamlessly incorporated into the backend of this pipeline to further extend its use and sensitivity.

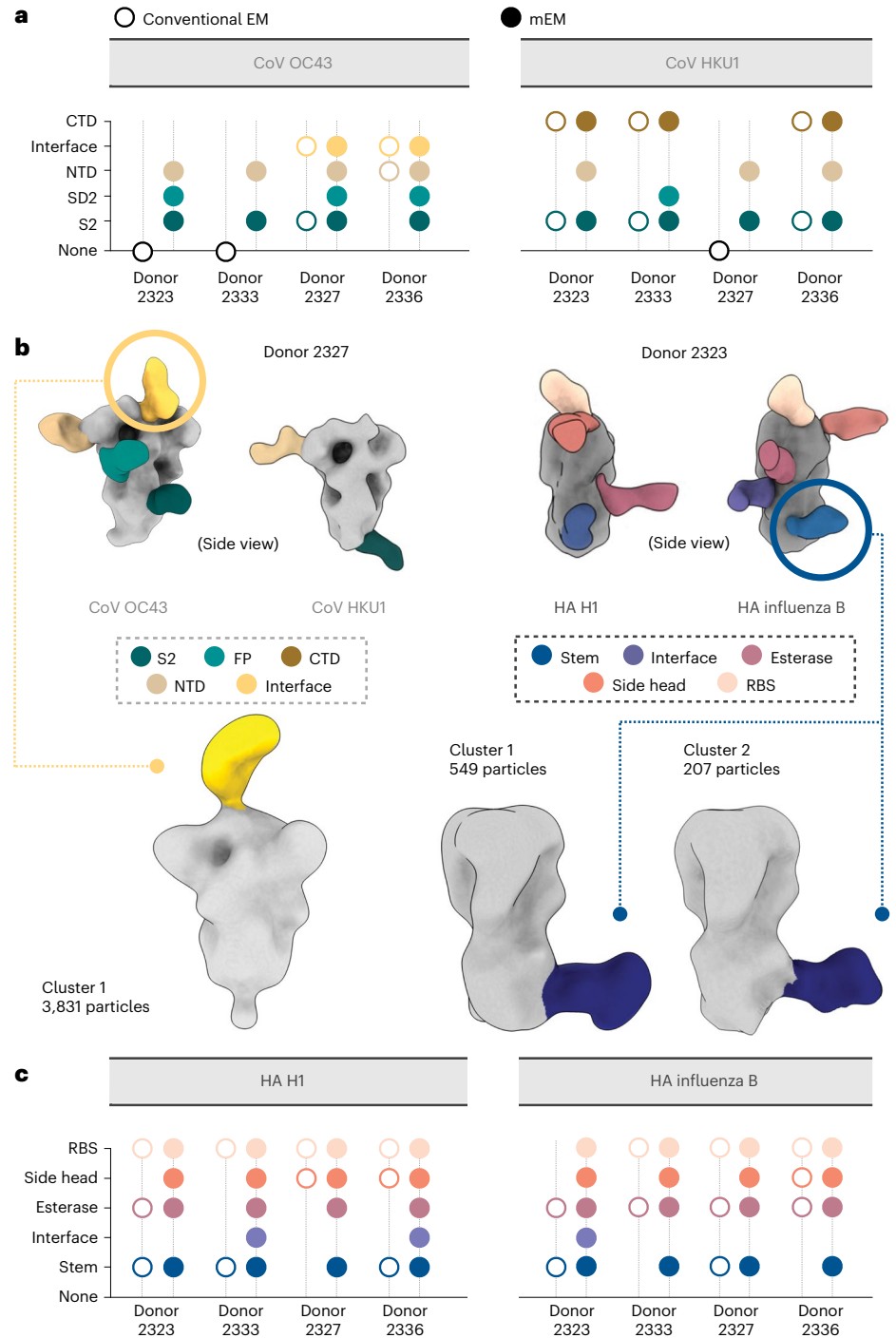

**Fig. 4 | Detection by mEM of immune responses from individuals vaccinated with Fluzone using ns-EM and cryo-EM. a**, Longitudinal polyclonal epitope mapping of preexisting responses against the CoV OC43 and HKU1 S. Convalescent sera from four patients participating in a Fluzone vaccination study were used for conventional EMPEM (open circles) and mEM (closed circles) analysis. Labels presented on the *y* axis correspond to epitopes on the S. **b**, Cryo-EMPEM analysis of polyclonal antibodies elicited by Fluzone against CoV OC43 S (left) and HA B/Maryland/15/16 (right). Immune complexes were generated using CoV OC43 S and plasma from donor 2327 at day 2 (left) and HA B/Maryland/15/16 complexed with purified polyclonal Fabs from donor 2323 at day 7 (right). The antigens are represented in grey, and Fab densities are coloured according to the legends in the centre of the panel (boxed). Epitope mapping by mEM using ns-EM is shown for both donors, and antibodies identified by cryo-EMPEM are indicated with coloured circles on the matched ns-EM composite maps. **c**, Longitudinal polyclonal epitope mapping of vaccine-elicited responses against two HA subtypes. Sera from the previous four patients were used to analyse conventional EMPEM versus mEM epitope characterization. Labels presented on the *y* axis correspond to epitopes, including receptor binding site (RBS), side head, esterase, interface and stem, on the HA glycoproteins.

mEM avoids the traditional bottlenecks of sample preparation, including large sera volumes, antibody cleavage, column purifications and extensive hands-on time and human supervision. We demonstrated the characterization of polyclonal antibodies bound to six different viral glycoproteins using less than 1% of previously required patient sera, 4 µl versus 500 µl, in a fraction of the sample preparation time, 90 min per sample compared with 5 days. mEM can now be applied to specimens sourced from biobanks or clinical studies where small

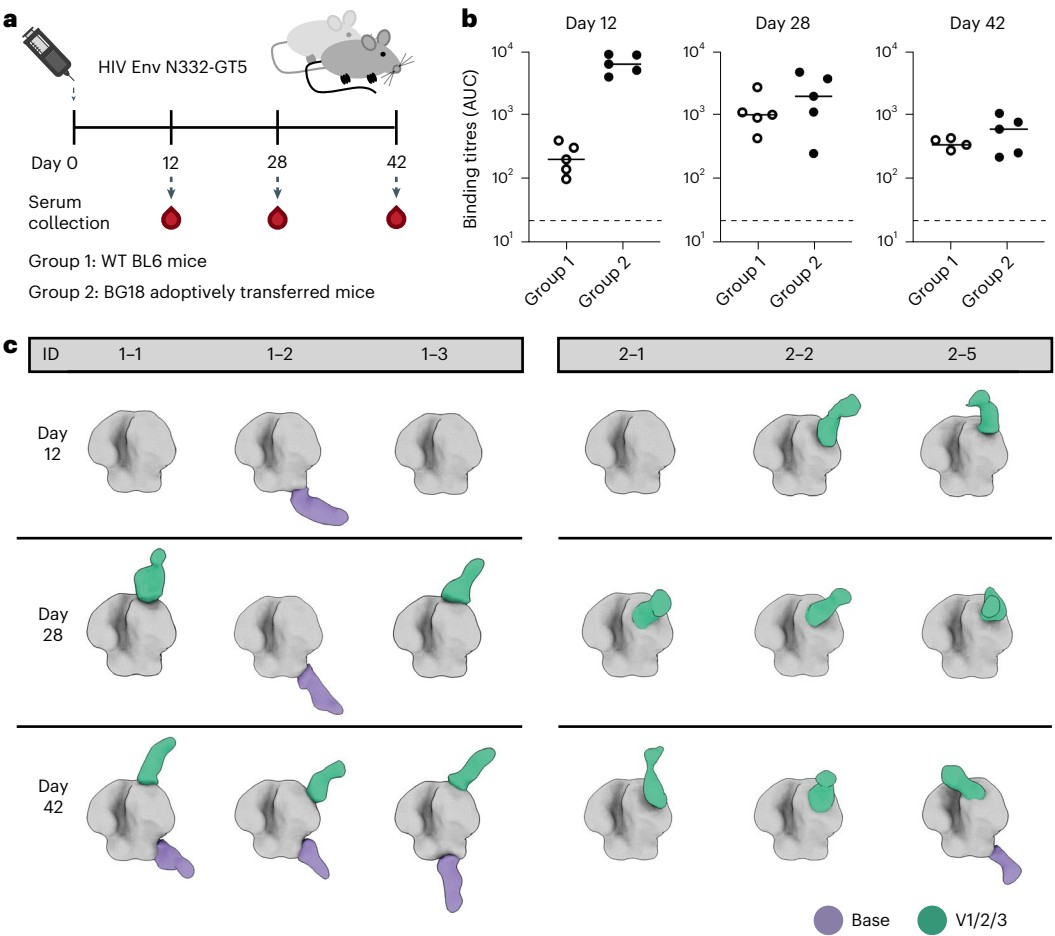

**Fig. 5 | Longitudinal analysis of immune responses from individual mice vaccinated with HIV Env N332-GT5 using mEM and ns-EM. a**, Immunization schedule. **b**, ELISA binding titres (area under the curve, AUC) for plasma samples collected at days 12, 28 and 42 (timepoints indicated above each graph; open circles: group 1, closed circles: group 2). Horizontal lines represent geometric mean values at each timepoint ($n = 5$ animals per group at each timepoint). ELISA experiments were performed in triplicates ($n = 3$) for each plasma sample, and the mean value is plotted. **c**, Composite figures from mEM analysis by ns-EM of polyclonal responses at days 12, 28 and 42. Animals with the highest ELISA titres ($n = 3$) in both groups were selected for epitope mapping, and the animal IDs for corresponding animals are shown above the day 12 composite figures. A colour-coding legend for antibodies targeting different epitope clusters is shown at the bottom right. HIV Env N332-GT5 glycoprotein is shown in grey.

volumes of annotated samples can be used to examine natural disease history, in early-stage vaccine trials where preclinical samples are limited, and paediatric vaccine trials where volumes of blood draws are much reduced compared with adult participants. With the mEM technology being both reusable and versatile, it is accessible for many research laboratories examining a variety of viral pathogens.

The ability to structurally assess immune complexes at a higher throughput compared with conventional EMPEM will enable epitope mapping of large-scale clinical studies with limited sample availability. We determined that polyclonal antibodies from individual mice could be observed longitudinally at early and intermediate timepoints in a vaccine trial, presenting a more complete picture of the epitopic landscape in mice early on despite minute sera volumes. We found that entire group responses ($N = 4$) could be complexed for characterization in less than a day, one-fifth of the time needed compared with conventional preparation methods, enabling real-time evaluation and candidate redesign. The application of mEM to an automated, multi-antigen system could further increase throughput and notably enable the parallelization of processing to produce larger datasets for immune complex determination, with the scale-up of collection and processing being an active area for optimization. Although lower-resolution characterization of immune complexes is readily achieved by mEM, improvements to increase glycoprotein concentration are necessary

for atomic-level assessment using cryo-EM. Moreover, the inherent flexibility and movement present with IgG binding, as well as the propensity of IgG to cross-link adjacent glycoproteins, require higher particle density to obtain high-resolution structures. Thus, the important work ahead will need to focus on the improved glycoprotein capture, immune complex aggregation and downstream imaging and processing workflows.

mEM resulted in improved sensitivity of antibody detection among almost all samples studied; additional epitopes on all six viral glycoproteins characterized were identified compared with conventional EMPEM methodology, including at the individual mouse level. The improved sensitivity could be attributed to multiple factors: (1) glycoprotein presentation by mEM potentially improves antibody capture; (2) conventional antibody purification and Fab preparation results in a loss of polyclonal antibodies, leaving mostly those that are highly abundant; (3) relatively long elution times and large volumes in conventional size exclusion chromatography result in dissociation of weaker affinity antibodies; and (4) mEM uses IgG, the bifunctionality of which improves avidity[39], enabling the capture of low-affinity or low-abundance antibodies otherwise lost. The higher sensitivity of mEM enabled the detection of a novel polyclonal antibody targeting the protomer–protomer site on both strains of HA in multiple donors, not previously observed by conventional EMPEM and recently described

as a non-neutralizing antibody present at low frequency in memory B cell populations[29].

mEM exhibits scalability, high sensitivity and reproducibility, making it well suited for the structural characterization of polyclonal antibody responses. The simplified approach and minute material requirements make mEM broadly accessible to vaccine (re)design workflows as well as for novel structure or antibody determination. We anticipate that, in creating a miniaturized technology for the structural evaluation of immune complexes, mEM will prove to be a valuable tool for studying protein–protein interactions by EM.

## Methods

### Protein purification and expression
For protein expression, FreeStyle 293F cells were transfected at a density of $1 \times 10^6$ cells ml$^{-1}$ as previously described[40]. In brief, PEI-MAX (1 mg ml$^{-1}$) was mixed with 250 µg of HIV glycoprotein and 62.5 µg of Furin plasmids at a 3:1 ratio (Env:Furin) in OPTI-MEM. For the other glycoproteins, Furin was not needed and 500 µg of glycoprotein was used. The cells and cultures were collected 6 days after transfection.

For HIV Env N355-GT5, glycoproteins were purified by affinity chromatography using a PGT145 column as previously described[41]. Supernatants were filtered through 0.45-µm filters and passed through the PGT145 column at a flow rate of 1 ml min$^{-1}$. Bound HIV Env glycoproteins were eluted using 1× column volume of 3 M MgCl$_2$ before buffer exchange to TN75 (75 mM NaCl and 10 mM Tris, pH 8.0). Trimers were further purified using a Superdex 200 increase 10/300 column (GE Healthcare Biosciences) in tris buffered saline (TBS buffer).

For HA and CoV glycoproteins, the proteins were purified from the supernatants using Strep-TactinXT 4FLOW gravity flow columns (IBA Lifesciences). The supernatant was passed through the columns at a flow rate of 1 ml min$^{-1}$. The protein was incubated with 1× BXT elution buffer (1 M Tris–Cl, 1.5 M NaCl, 10 mM EDTA and 50 mMM biotin, IBA Lifesciences) overnight, eluted and further purified in TBS buffer using a Superdex 200 increase 10/300 column and a Superose 6 increase 10/300 column (GE Healthcare Biosciences), for HA and CoV glycoproteins, respectively.

All the protein fractions corresponding to the trimeric proteins were collected and concentrated using a 30 kDa cut-off Amicon ultrafiltration unit. The quality of the proteins was assessed by ns-EM for further use.

### Surface plasmon resonance
All SPR-based assays were performed on a Biacore T200 instrument at 25 °C.

**Gold surface functionalization.** For the MHDA surface, gold sensor chips were cleaned by sonication for 2 min in acetone, ethanol and MilliQ water (see 'Microfluidic flow cell fabrication' section). Next, the chips were incubated in 5 mM MHDA (Sigma) dissolved in ethanol in dark conditions at room temperature (RT) for 4 days. The chips were then rinsed in ethanol and dried with nitrogen and stored at 4 °C in dark conditions until use.

**Strep-TactinXT covalent immobilization.** Strep-TactinXT (IBA Lifesciences) was immobilized at three different concentrations (0, 50 and 100 µg ml$^{-1}$) via standard 1-ethyl-3-(3-dimethylaminopropyl-carbodiimide hydrochloride (EDC)– N-hydroxy-succinimide (NHS) amine coupling chemistry, followed by ethanolamine to stop the coupling reaction. Immobilization was carried out in situ with a flow rate of 15 µl min$^{-1}$ in a running buffer of deionized water (degassed) pH 7.

**Spike glycoprotein capture, Ab–Ag complex formation and elution.** All experiments were carried out with a flow rate of 15 µl min$^{-1}$ in a running buffer of degassed 1× phosphate-buffered saline (PBS) + Tween-20 pH 7.4. A concentration series of CoV spike glycoproteins containing a Twin-Strep-tag were injected over the Strep-TactinXT surface for 2 min, followed by a 2-min dissociation phase. Regeneration of the surface in between injections of spike glycoprotein was achieved by first injecting 50 mM biotin (IBA Lifesciences) for 1 min followed by three 120-s injections of 3 M GuHCl. Kinetic analysis of each reference subtracted injection series was performed using the BIAEvaluation software (Cytiva) and GraphPad Prism.

### Microfluidic flow cell fabrication
**PDMS cell.** The microfluidic cell was fabricated using a standard soft lithography technique. Autocad was used for the designed pattern consisting of four 20-µl channels (60 mm × 1.6 mm × 0.06 mm), which was then transferred onto a silicon wafer coated with a photoresist (SU-8) to create a master mould (UCSD Nano3 Facility). The master was examined by digital microscopy to verify structure details. Next, the microchannels were formed in PDMS (Sigma-Aldrich), which was mixed in a 10:1 ratio of polymer base to curing agent. The mixture was poured over the master mould (3 mm thickness) and degassed in a desiccator for approximately 1 h or until all air bubbles were removed. The curing was then finalized by placing the polymer mould in an oven at 75 °C for 2 h, after which the PDMS was detached from the master and access holes for tubing were punched using a biopsy tool of 1 mm diameter (Fisher Scientific). In addition, the PDMS cell was cut with a 75 mm × 25 mm border around the microchannels to match the dimensions of a standard microscope slide (Fisher Scientific). Finally, the polymer device was exposed to oxygen plasma in plasma cleaner Solarius (Gatan) at a process pressure of 1,000 torr, a power of 20 W, a flow rate of 40 sccm and a duration of 2.5 min. The PDMS was then immediately coated with 200-PEG (Sigma-Aldrich) mixed at a 1:1 ratio with MilliQ water and placed in an oven at 250 °C for 25 min to make the microchannels hydrophilic. The remaining PEG was then removed with subsequent washing in ethanol and water, and the PDMS cell was dried with N$_2$. The PDMS cell was kept at RT for an extended time with no detriment to the surface blocking or microchannels (>1 year).

**Biofunctional SAM surface.** Gold-coated microscope slides (50 nm thickness, Electron Microscopy Sciences (EMS)) were used to form a biofunctional SAM surface. Slides were cleaned with acetone, ethanol and MilliQ water by sonication at 60 °C for 5 min and subsequently dried with nitrogen before exposure to oxygen plasma in plasma cleaner Solarius (Gatan) at a process pressure of 1,000 torr, a power of 20 W, a flow rate of 40 sccm and a duration of 15 min. A SAM was created with MHDA (Sigma) solubilized in 200-proof ethanol (Sigma) at a concentration of 5 mM. Gold slides were incubated in 15 ml of the MHDA solution in dark conditions at RT for 4 days, rinsed with 200-proof ethanol and dried with nitrogen. Slides could be stored in 200-proof ethanol for at least 2 weeks before use. Amine coupling (1:1 v/v 0.2 M EDC and 0.05 M NHS, Cytiva) for 10 min at RT was used to attach Strep-TactinXT (IBA Lifesciences), which was dropcast onto the slide at 100 µg ml$^{-1}$ (sodium acetate pH 4.5) for 1 h. Finally, the gold slides were thoroughly rinsed with MilliQ water and dried with nitrogen.

**PMMA clamp.** The biofunctional flow cell was assembled using a specially designed clamp composed of clear poly(methyl methacrylate) (PMMA) cut into two pieces sized 50 mm × 76 mm. The top piece contains 1-mm-diameter tubing holes corresponding to the PDMS cell. The cell is then placed on top of the biofunctional gold slide and clamped between the two PMMA pieces. Standard 4 mM screws are used to tighten the components together and form a leak-proof seal. The use of a removable PMMA clamp to form the biofunctional flow cell enables the system to be entirely reusable and recyclable.

### mEM system set-up and operation
To set up the mEM system, the different buffers and reagents are delivered to the clamped biofunctional flow cell using polytetrafluorethylene

(PTFE, VWR) tubing (0.76 mm inner diameter), which is connected to the inlet and outlet ports of the PDMS cell, a syringe pump (NE300, New Era), six-manifold injection valve (V451, Idex) and a sample collection tube (0.6-ml Eppendorf tube). The final volume of the microfluidic flow path is 56 or 72 µl depending on the type of EM analysis (ns- or cryo-EM) and composed of the following sections: 20 µl sample loop, 10 or 20 µl valve-to-chip connection, 20 µl microchannel and 6 or 12 µl exit tube. As the system running buffer, 1× TBS is used. Samples are injected into the sample loop and then loaded through the flow path at either 5 µl min⁻¹ or 15 µl min⁻¹, and protein flow and elution times are calculated accordingly (Supplementary Fig. 1). Finally, EM is used to evaluate the final elution fraction either by heavy-metal staining of the sample with uranyl formate or sample vitrification in ice. The sample injection loop is cleaned with 70% ethanol and MilliQ water between each solution addition. All PTFE-connected components of the microfluidic set-up are reused after cleaning with 70% 2-isopropanol followed by MilliQ water once sample collection is completed.

### Glycoprotein capture by mEM

To determine optimal glycoprotein concentration for immobilization, SARS-CoV-2 S was diluted in TBS to the following concentrations: 0.1 mg ml⁻¹, 0.5 mg ml⁻¹, 1 mg ml⁻¹, 2 mg ml⁻¹ and 5 mg ml⁻¹. In addition, CoV OC43, CoV HKU1, HA H1, HA influenza B and HIV Env N332-GT5 were diluted to 2 mg ml⁻¹ in TBS. For each SARS-CoV-2 concentration, 20 µg was injected into the sample loop and then loaded over the flow cell at 5 µl min⁻¹ for 14.5 min. Unbound glycoprotein was washed from the channel at 15 µl min⁻¹ for 20 min. To elute, 20 µl of 1× BXT was injected into the sample loop and loaded into the channel at 5 µl min⁻¹. After 4 min, the elution was collected in a 0.6-ml Eppendorf tube for 10.5 min, with 3 µl directly placed on a 400-mesh Cu grid for staining by uranyl formate (UF) or plunge-frozen for evaluation by EM. Nanodrop was used to determine elution concentration. The same protocol was followed for each additional glycoprotein, and the shorter tubing connections described previously for the cryo-EM mEM system set-up were used (see 'mEM system set-up and operation' section).

### Antibody complexing with mEM

For monoclonal antibody complexing, 3 molar excess of TXG-0078 Fab and CC6.30.2 IgG to SARS-CoV-2 S was calculated and both monoclonal antibodies were diluted into TBS accordingly. For polyclonal antibody complexing, polyclonal IgG and Fab were purified as described. Based on the glycoprotein elution concentration determined using Nanodrop, we assumed that roughly 2 µg of protein was immobilized on the Strep-TactinXT surface. Thus, to maintain the conventional EMPEM complexing ratio (10 µg glycoprotein:330 µg polyclonal Fab), 66 µg of purified polyclonal Fab and 22 µg of purified polyclonal IgG were used for complexing with SARS-CoV-2 spike glycoproteins. Less polyclonal IgG was used compared with Fab to reduce the chances of cross-linking and glycoprotein–antibody aggregation. Next, 20 µl of SARS-CoV-2 spike at 1 mg ml⁻¹ was injected into the sample loop and then loaded over the flow cell at 5 µl min⁻¹ for 14.5 min. Unbound glycoprotein was washed from the channel at 15 µl min⁻¹ for 20 min. Next, 20 µl of monoclonal or polyclonal antibodies were injected into the sample loop and loaded into the channel at 5 µl min⁻¹ for 14.5 min. Unbound antibodies were washed from the channel at 15 µl min⁻¹ for 30 min. To elute the antibody–glycoprotein complexes, 20 µl of 1× BXT was injected into the sample loop and loaded into the channel at 5 µl min⁻¹. After 4 min, the elution was collected in a 0.6-ml Eppendorf tube for 10.5 min, with 3 µl immediately placed on a 400-mesh Cu grid and stained with UF for evaluation by ns-EM.

### Polyclonal antibody isolation from sera using mEM

All serum samples were heat-inactivated at 65 °C for 1 h before use. First, 4 µl (1:10) of donor 74-4 plasma was diluted into 40 µl of TBS. Next, 20 µg of SARS-CoV-2 S at 1 mg ml⁻¹ was injected into the sample loop

and passed through the microfluidic channel at 5 µl min⁻¹ for 14.5 min. Unbound glycoprotein was washed from the channel at 15 µl min⁻¹ for 20 min. Then, 5% BSA was injected into the sample loop and loaded into the flow cell at 15 µl min⁻¹ for 5 min to prevent non-specific protein interactions from the plasma and serum samples. The channel was washed with TBS for 5 min at 15 µl min⁻¹ to remove excess BSA. Twenty microlitres of diluted plasma was injected into the sample loop and loaded into the channel at 5 µl min⁻¹ for 14.5 min. Unbound antibodies were washed from the channel at 15 µl min⁻¹ for 35 min. To elute the antibody–glycoprotein complexes, 20 µl of 1× BXT was injected into the sample loop and loaded into the channel at 5 µl min⁻¹. After 4 min, the elution was collected in a 0.6-ml Eppendorf tube for 10.5 min, with 3 µl immediately placed on a 400-mesh Cu grid for staining by UF. Next, all Twin-Strep-tagged glycoprotein, including CoV OC43, CoV HKU1, HA H1, HA influenza B and HIV Env N332-GT5, were prepared at a concentration 1 mg ml⁻¹. mEM samples from all donors were diluted 1:10 with 4 µl of plasma added to 40 µl TBS, with the exception of the BG18 mice samples which were diluted 1:100 with 0.4 µl of plasma added to 39.6 µl TBS. The same protocol was followed for each donor sample complexed with the corresponding glycoproteins.

For sample evaluation by cryo-EM, the mEM system was set up with shorter PTFE tubing connections to decrease sample elution volume (see 'mEM system set-up and operation' section). Next, CoV OC43 S at 1 mg ml⁻¹ was injected into the sample loop twice and loaded into the microfluidic channels at 5 µl min⁻¹ for 16.5 min. Specifically, 20 µl was added to the loop and loaded into the microchannel for 4 min to allow the loop to fully empty. Next, an additional 20 µl was injected into the loop and then loaded into the system for an additional 11 min. This was done to increase the concentration of immobilized glycoprotein on the Strep-TactinXT surface. Unbound glycoprotein was washed from the channel at 15 µl min⁻¹ for 20 min. BSA (5%) was then injected into the sample loop and loaded into the flow cell at 15 µl min⁻¹ for 5 min to prevent non-specific protein interactions, after which the channel was washed with TBS for 5 min at 15 µl min⁻¹. Next, 20 µl of diluted plasma was then injected into the sample loop and loaded into the channel at 5 µl min⁻¹ for 11 min. Unbound antibodies were washed from the channel at 15 µl min⁻¹ for 35 min. To elute the antibody–glycoprotein complexes, 20 µl of 1× BXT was injected into the sample loop and loaded into the channel at 5 µl min⁻¹. After 4 min, the elution was collected in a 0.6-ml Eppendorf tube for 7 min and immediately plunge-frozen for evaluation by cryo-EM.

### HAI

Serum samples were collected before vaccination (day 0) and after vaccination (day 28). HAI assays were performed as previously described[42]. In brief, serum samples for immunogenicity evaluation were assessed by HAI assays against egg-derived vaccine A/H1N1, A/H3N2 and B-strain antigens. Serum samples were pretreated with receptor destroying enzyme to block non-specific inhibitors and tested at an initial dilution of 1:10. The HAI titre was the highest dilution that showed complete inhibition of haemagglutination.

### ELISA

Microlon-600 96-well half-area plates (Greiner Bio-One) were coated for 1 h with purified HA at 25 µg ml⁻¹ in 0.1 M NaHCO₃, pH 8.6 (50 µl per well). Unbound trimers were removed by three wash steps with TBS 0.1% Tween-20 before before blocking with PBS 5% BSA overnight. For HIV, Pierce streptavidin-coated plates were coated for 1 h with HIV Env at 3.5 µg ml⁻¹ in 0.1 M NaHCO₃, pH 8.6 (100 µl per well). Unbound trimers were removed by three wash steps with TBS 0.1% Tween-20 before serially diluting the sera in threefold steps starting at 1:30 dilution. After three washes with TBS 0.1% Tween-20, AP-conjugated AffiniPure goat anti-human IgG for HA and AP-conjugated goat anti-mice IgG for HIV (Jackson Immunoresearch, cat. no. 109-055-097) was added at a 1:5,000 dilution in TBS 1% BSA. Colorimetric detection was performed using

alkaline phosphatase (pNPP) liquid substrate (Thermo-Fisher Scientific). Colour development (absorption at 405 nm) was stopped using 2 M NaOH (25 µl) when a plateau value was reached in the first two wells containing the highest sera concentration (for HA) or background was above 0.25 (for HIV). Data were recorded on a Synergy H1 plate reader (BioTek), and curves and midpoint titres were plotted and calculated using Prism version 8.3.0. Experiments were performed in duplicate. Data are represented as mean ± s.e.m. using Prism.

### Serum IgG isolation and antibody digestion for HA using conventional method

IgG from human sera was isolated using CaptureSelect IgG-Fc Affinity Matrix (Thermo Scientific) as previously described[43]. In summary, 0.5 ml of human sera was mixed with 0.5 ml of washed CaptureSelect resin and 4 ml of PBS. For IgG digestion, the unbound IgG was discarded and the resin was kept. Papain was activated for 15 min at 37 °C in digestion buffer (100 mM Tris, 2 mM EDTA, 10 mM L-cysteine and 1 mg ml$^{-1}$ papain) and was added to the resin containing the IgG. Subsequently, digestion buffer was added to the resin up to a total volume of 5 ml and the mixture was incubated for 4–5 h at 37 °C. Iodocetamide was used to quench the reaction at a final concentration of 0.03 M, and the Fab/Fc was purified and concentrated using size exclusion chromatography (Superdex 200 Increase column, Cytiva Life Sciences). The fractions containing purified Fabs/Fc were concentrated using 10 kDa Amicon ultrafiltration units.

For IgG-specific mEM assays, IgG from human sera was isolated using a 6-ml packed column composed of CaptureSelect IgG-Fc Affinity Matrix (Thermo Scientific). All sera were first heat-inactivated at 56 °C for 1 h. The sera were then spun at 1,500$g$ for 5 min, filtered with a 0.2-µm centrifuge filter unit and diluted up to 2 ml using 1× PBS. Tween-20 detergent was added to each sample for a 1% final concentration, mixed until the detergent was fully dissolved and then incubated for 30 min before being transferred to a 96-deep-well plate. An ALIAS autosampler was used to inject each sample over a 6-ml CaptureSelect IgG-Fc column on an AKTA Pure Protein Purification System (Cytiva). Fractions corresponding to IgG peaks were collected and buffer exchanged into 1× TBS.

### Purification of glycoprotein–polyclonal Fab complexes

Glycoprotein–polyclonal Fab complexes were generated by incubating 10–15 µg of glycoprotein with 0.37–0.5 mg of polyclonal Fab overnight at RT. The complexes were purified using a Superdex 200 increase column on Akta Pure system (GE Healthcare) running in TBS buffer. The fractions containing the complexes were concentrated using 10 kDa amicon ultrafiltration units and immediately added to a negative-stain grid.

### Negative-stain data processing

For all CoV and HA glycoprotein–antibody complexes, ~100,000 particles were picked using Appion image processing package. Particles were transferred to Relion 3.0 and two-dimensional (2D) classification was performed[44]. Particles that contained trimer only or trimer–antibody complexes were selected for further 3D analysis. The 3D reference models for 3D classifications and refinements were a low-resolution model of a non-liganded CoV and HA, respectively. For all glycoprotein–antibody complexes, initial 3D refinement was performed after 2D classification using the selected particle stack to align all the particles before 3D classification. Particles were then classified into 8–60+ classes depending on overall particle number and to ensure a minimum of 1,000 particles per class. Three-dimensional classes with similar features were combined and refined. All maps with clear antibody densities were visualized and segmented using UCSF Chimera, and composite images were produced in ChimeraX[45].

For HIV Env glycoprotein–antibody complexes (Supplementary Fig. 10b), ~200,000 particles were picked using Relion 4.0 (ref. [46])

and three rounds of subsequent 2D classification were performed. Particles that contained trimer only or trimer–antibody complexes were selected for 3D focused classification where the selected particles are reconstructed onto an apo trimer map, C3 symmetry expansion is applied and seven masks for broad epitopes (base, gp41 glycan hole, gp41 fusion peptide, gp120 glycan hole, CD4bs–gp120 interface, C3V5 and V1V2V3) are automatically generated. For all seven epitopes, ChimeraX[45] was then used to assess the density at each (contour level >0.02), with maps that contain Fab-like densities either combined (if similar) or grouped separately. A round of 3D refinement is run for all selected maps followed by another round of 3D focused classification for each selected epitope. This step is repeated two to three times until there is convergence for the 3D refinement maps for epitopes with Fab densities. All maps with clear antibodies densities were visualized and segmented, and composite images were produced in ChimeraX[45].

### Cryo-EM sample preparation

Protein, specifically, apo CoV OC43, Apo HA B/Maryland/15/16, Apo SARS-CoV-2, CoV OC43 complexed with 182327 sera at day 7 and HA B/Maryland/15/16 complexed with 182323 purified Fab at day 7, eluted from the mEM platform was deposited on a graphene oxide on holey carbon copper mesh grids (Electron Microscopy Sciences). Specifically, samples were double blotted with 4 µl of sample and an interval of 30 s in between sample deposition. After the blot step, the grids were plunge-frozen into liquid-nitrogen-cooled liquid ethane. A Vitrobot Mark IV (Thermo Fisher) set to 4 °C, 100% humidity, 30 s wait time and 2.5 s and 3.5 s blot time for CoV and HA, respectively, was used.

### Cryo-EM data collection and image processing

For apo CoV OC43, HA B/Maryland/15/16 and SARS-CoV-2, data were collected using Leginon[47] on a Thermo Fisher Arctica operating at 200 keV with a Gatan K2 Summit direct electron detector. Micrographs were aligned and dose weighted using MotionCor2[48]. Micrographs were imported to cryoSPARC v3.2 (ref. [49]), contrast transfer function (CTF) was estimated using GCTF[50], and the micrographs were denoised using Topaz denoise[51]. Particle picking was done by first using an automated blob picker with a radius of 200 Å, followed by automated picking using templates created from an initial 2D classification, and the last round of picking was done using Topaz[52]. Particles were extracted and subjected to a few rounds of 2D classification for cleaning. An ab initio model was generated, and several rounds of homogeneous and heterogeneous refinement were performed. Global and local CTF refinement, 3D variability analysis and non-uniform refinement[53] were performed with C3 symmetry applied, which resulted in a final reconstruction of 3.0 Å for OC43 S and 3.3 Å for influenza B HA (Fourier shell correlation, $FSC_{0.143}$) (Supplementary Table 1). For SARS-CoV-2, additionally, local refinement and 3D variability analysis using a mask on the 'head' of the protein was performed to generate the two states of the protein, final reconstructions of 4.6 Å ('down') and 6.9 Å ('up') ($FSC_{0.143}$).

For cryo-EMPEM datasets, CoV OC43 bound to IgG from 182327 day 7 sera and HA B/Maryland/15/16 bound to Fabs from 182323 day 7, data were collected using Thermo Scientific Smart EPU Software[54] on a Thermo Fisher Glacios 2 operating at 200 keV with a Thermo Fisher Falcon IV direct electron detector. Micrographs were aligned and dose-weighted using cryoSPARC Live and imported to cryoSPARC. A small dataset consisting of 250–500 micrographs of apo CoV OC43 and apo HA B/Maryland/15/16 was collected to do Topaz training on the dataset and extrapolate the Topaz training model to the cryo-EMPEM datasets. To process the cryo-EMPEM datasets, particles were picked using the Topaz-trained model on apo particles. Particles were subjected to a few rounds of 2D classification for cleaning, and a Topaz model was trained on the cleaned particles. Particles were picked, extracted and subjected to a few more rounds of 2D classification for cleaning. Several rounds of homogeneous and heterogeneous refinement were performed, and particles were

exported to Relion 4.0 (ref. 44). A 3D refinement was performed, and low-pass-filtered maps of the apo trimers were used as initial models for all 3D steps to avoid initial model bias for the Fabs. Particles were symmetry-expanded using C3 to collapse all the epitope interfaces onto a single protomer. To prevent copies of the individual particles to align to themselves, particle alignment was constraint in the subsequence 3D classification and refinement steps (--skip_align - T = 16 and local angular searches of 3.7° per iteration for 3D classification and 3D refinement, respectively). The first round of 3D classification was run with a 80 Å or 60 Å sphere mask around the epitope interface for IgG and Fab, respectively. The number of 3D classes was adjusted for each dataset and occupancy, and a range between 10 and 120 classes was used. Particles from the 3D classes that showed Fabs on the epitope–paratope region were selected for a round of 3D refinement. Other rounds of 3D classification were run using sphere masks around the epitope until the number of particles was too small to sort more classes. The final round of 3D classification was run using a full trimer-Fab mask, and the final class selected was refined and postprocessed using MTF correction.

### Model building and refinement
Initial model for CoV OC43 was generated using the PDB 6OHW and fitted into the refined cryo-EM map. Three rounds of manual and automated model building and relaxed refinement were performed using Coot v0.9.8 (ref. 55) and Phenix[56]. Validation of the models as done using EMRinger and MolProbity in Phenix[56]. Final statistics and PDB/EMDB deposition codes are presented in Supplementary Table 1.

### Human and mouse samples used in the study
**Human CoV study details.** Plasma samples from two donors (Lotus 74-4 and Lotus 78-4) were used for EMPEM studies. Deidentified convalescent plasma was provided through the 'Collection of Biospecimens from Persons Under Investigation for 2019-Novel Coronavirus Infection to Understand Viral Shedding and Immune Response Study' UCSD institutional review board number 200236. The protocol was approved by the UCSD Human Research Protection Program. Samples were collected on the basis of COVID-19 diagnosis through a rapid antigen test or PCR. All samples were obtained after written informed consent.

**Human influenza study details.** Plasma samples from four donors (182323, 182333, 182327 and 182336) were used for EMPEM. Subjects were part of the study 0409027018 from Yale University. In brief, healthy participants were immunized with high-dose trivalent Fluzone provided by Sanofi Pasteur. All participants provided informed consent. Serum was collected at days 0, 2, 7, 28 and 70 after immunization at day 0.

**Mouse HIV study details.** Plasma samples from 30 mice (WT #1-5 and BG18 #1-5) were used for mEM studies. Animal samples have been described previously[23]. Healthy adult male and female C57BL/6J (CD45.2+/+) mice heterozygous for the BG18gH knock-in, generated as previously reported[32], and 8–12-week-old male B6.SJL-Ptprca Pepcb/BoyJ mice (CD45.1+/+) purchased from The Jackson Laboratory were used in this study. Mice were housed at the animal facility, with free access to food and water, controlled temperature and a 12:12 h light–dark cycle. Mice were not involved in previous procedures and were drug and test naive. The mouse maintenance and experiments were performed following the approved protocols by the Institutional Animal Care and Use Committee of Massachusetts General Hospital, an Association for Assessment and Accreditation of Laboratory Animal Care International-accredited facility, under animal study protocols 2016N000286 and 2016N000022.

For all the assays described in the Article, plasma samples for Lotus donors 74-4 and 78-4 were used for SARS-CoV-2 spike glycoproteins, plasma samples for donors 182323, 182333, 182327 and 182336

were used to test CoV OC43, CoV HKU1, HA H1 and HA influenza B, and plasma samples for mice (WT #1-3 and BG18 #1-2, 5) were used for HIV Env N332-GT5 glycoprotein and N332-GT5 mRNA for preparing the mouse serum samples.

### Reporting summary
Further information on research design is available in the Nature Portfolio Reporting Summary linked to this article.

## Data availability
Three-dimensional maps and models for the EM analysis have been deposited to the Electron Microscopy Databank (EMDB) (http://www.emdatabank.org/) and Protein Data Bank (PDB) (http://www.rcsb.org/), respectively (EMDB IDs: 44655-65, 44667-70, 44679-80 and 44682-83; PDB IDs: 9BLK and 9BTO). Final statistics and PDB and EMDB deposition codes are presented in Supplementary Table 1, and all ns-EM EMDB accession numbers are listed in Supplementary Table 2. This Article does not report original code. Any additional information required to reanalyse the data reported in this Article, including raw micrographs and/or particle stacks, is available from the lead contact upon request.

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

## Acknowledgements

We thank B. Anderson for engineering and microfluidic system support. We thank W. Lee, H. L. Turner, A. Gharpure and W. Lessen for EM data collection support and C. Bowman and J. C. Ducom for computational support. We thank the Scripps Research Institute Biophysics Core for SPR technical support. This work was supported by NIH NIAID grants AI136621 (A.B.W.), AI089992 (A.B.W. and A.C.S.) and AI144462 (A.B.W. and D.R.B.) and by the Bill and Melinda Gates Foundation INV-002916 (A.B.W.). This work was performed in part at the San Diego Nanotechnology Infrastructure (SDNI) of UCSD, a member of the National Nanotechnology Coordinated Infrastructure, which is supported by the National Science Foundation (grant number ECCS-2025752). We are thankful for the Netherlands Organization for Scientific Research (NWO) Rubicon Grant 45219118 to A.T.d.l.P. and to the Scripps Research Institute

Skaggs Graduate School for the David C. Fairchild Endowed Fellowship to L.M.S.

## Author contributions

L.M.S., A.T.d.l.P., B.C.R. and A.B.W. conceived the microfluidic technology. L.M.S. and A.T.d.l.P. performed optimization of the mEM protocol for the several glycoproteins and monoclonal and polyclonal samples as well as ELISA, SPR, ns-EM and cryo-EM experiments. R.d.P.F.R. performed influenza EMPEM, ELISA and ns-EM experiments. G.G. performed protein purification, ns-EM imaging and data processing. M.L. carried out experiments for CoV EMPEM. A.S.T. processed serum samples. L.M.S., A.T.d.l.P., B.C.R., S.B. and G.O. analysed data. Z.X. and F.D.B. performed HIV N332-GT5 immunization study and provided mouse plasma samples. S.M. and A.C.S. provided influenza serum samples and HAI titres. N.B., T.F.R. and D.R.B. provided SARS-2 plasma samples and neutralization titres. L.M.S., A.T.d.l.P., B.C.R. and A.B.W. wrote the paper. A.B.W. supervised the research. All authors contributed to the manuscript text by assisting in writing or providing feedback.

## Competing interests

A.B.W. is an inventor on US patent 11217328 describing the implementation of EMPEM for epitope mapping. L.M.S., A.T.d.l.P., B.C.R. and A.B.W. are listed as inventors on a US (provisional) patent application filed 8 January 2025 describing the use of a microfluidic device for EMPEM. The other authors declare no completing interests.

## Additional information

**Correspondence and requests for materials** should be addressed to Andrew B. Ward.

[1]Department of Integrative Structural and Computational Biology, The Scripps Research Institute, La Jolla, CA, USA. [2]Department of Surgery, University of California San Francisco School of Medicine, San Francisco, CA, USA. [3]Ragon Institute of Mass General, MIT and Harvard, Cambridge, MA, USA. [4]Department of Medicine, Section of Infectious Diseases, Yale University School of Medicine, New Haven, CT, USA. [5]Department of Immunology and Microbiology, The Scripps Research Institute, La Jolla, CA, USA. [6]Division of Infectious Diseases, Department of Medicine, University of California, San Diego, La Jolla, CA, USA. [7]Department of Biology, Massachusetts Institute of Technology, Cambridge, MA, USA. ✉e-mail: andrew@scripps.edu

# Reporting Summary

## Statistics

For all statistical analyses, confirm that the following items are present in the figure legend, table legend, main text, or Methods section.

| n/a | Confirmed | |
|---|---|---|
| ☐ | ☒ | The exact sample size (*n*) for each experimental group/condition, given as a discrete number and unit of measurement |
| ☐ | ☒ | A statement on whether measurements were taken from distinct samples or whether the same sample was measured repeatedly |
| ☒ | ☐ | The statistical test(s) used AND whether they are one- or two-sided *Only common tests should be described solely by name; describe more complex techniques in the Methods section.* |
| ☒ | ☐ | A description of all covariates tested |
| ☒ | ☐ | A description of any assumptions or corrections, such as tests of normality and adjustment for multiple comparisons |
| ☐ | ☒ | A full description of the statistical parameters including central tendency (e.g. means) or other basic estimates (e.g. regression coefficient) AND variation (e.g. standard deviation) or associated estimates of uncertainty (e.g. confidence intervals) |
| ☒ | ☐ | For null hypothesis testing, the test statistic (e.g. *F*, *t*, *r*) with confidence intervals, effect sizes, degrees of freedom and *P* value noted *Give P values as exact values whenever suitable.* |
| ☒ | ☐ | For Bayesian analysis, information on the choice of priors and Markov chain Monte Carlo settings |
| ☒ | ☐ | For hierarchical and complex designs, identification of the appropriate level for tests and full reporting of outcomes |
| ☒ | ☐ | Estimates of effect sizes (e.g. Cohen's *d*, Pearson's *r*), indicating how they were calculated |

*Our web collection on statistics for biologists contains articles on many of the points above.*

## Software and code

Policy information about availability of computer code

| Data collection | Leginon (beta version), EPU (ThermoFisher) |
|---|---|
| Data analysis | Relion (v3.0 & 4.0), CryoSPARC (v2.15), Octet System Data Analysis (v9.0), Excel (v16.43), GraphPad Prism (v8.4.3), Appion (v1), UCSF Chimera (v1.13), MotionCor (v2), MolProbity (v4.2), EMRinger (version N/A), Localized Reconstruction (v1.2.0), GCTF (v1.06_sm_30_cu8.0_x86_64), Coot (v0.9-pre), BIAEvaluation (1.1.1) |

For manuscripts utilizing custom algorithms or software that are central to the research but not yet described in published literature, software must be made available to editors and reviewers. We strongly encourage code deposition in a community repository (e.g. GitHub). See the Nature Portfolio guidelines for submitting code & software for further information.

## Data

Policy information about availability of data

All manuscripts must include a data availability statement. This statement should provide the following information, where applicable:
- Accession codes, unique identifiers, or web links for publicly available datasets
- A description of any restrictions on data availability
- For clinical datasets or third party data, please ensure that the statement adheres to our policy

3D maps and models for the EM analysis have been deposited to the Electron Microscopy Databank (EMDB) (http://www.emdatabank.org/) and Protein Data Bank (PDB) (http://www.rcsb.org/), respectively. EMDB IDs: 44655-65, 44667-70, 44679-80, 44682-83. PDB IDs: 9BLK and 9BTO. Final statistics and PDB/EMDB deposition

codes are stated in Table S1 and all ns-EM EMDB accession numbers are listed in Table S2. This paper does not report original code. Any additional information required to reanalyze the data reported in this paper, including raw micrographs and/or particle stacks, is available from the lead contact upon request.

# Research involving human participants, their data, or biological material

Policy information about studies with human participants or human data. See also policy information about sex, gender (identity/presentation), and sexual orientation and race, ethnicity and racism.

| | |
|---|---|
| Reporting on sex and gender | n/a |
| Reporting on race, ethnicity, or other socially relevant groupings | n/a |
| Population characteristics | n/a |
| Recruitment | n/a |
| Ethics oversight | De-identified plasma/sera samples were obtained from UCSD and Yale University as described in the manuscript. Samples were shared for secondary research purposes as permitted by the Informed Consents used upon study enrollment. |

Note that full information on the approval of the study protocol must also be provided in the manuscript.

# Field-specific reporting

Please select the one below that is the best fit for your research. If you are not sure, read the appropriate sections before making your selection.

☒ Life sciences ☐ Behavioural & social sciences ☐ Ecological, evolutionary & environmental sciences

For a reference copy of the document with all sections, see nature.com/documents/nr-reporting-summary-flat.pdf

# Life sciences study design

All studies must disclose on these points even when the disclosure is negative.

| | |
|---|---|
| Sample size | Sizes of sample groups were minimized for sample availability concerns but still provide statistical significance. For ELISA and HAI assays, the experiments were performed in triplicates (n=3) which is a field-accepted standard for these types of experiments. |
| Data exclusions | No data was excluded for analysis. |
| Replication | ELISA, HAI and glycoprotein immobilization experiments were repeated in triplicates (n=3). All EMPEM experiments were performed once (n=1). No Data was excluded. |
| Randomization | Patient samples were selected in a randomized manner based on serum/plasma availability. Randomization is not applicable to the experiments performed in this study since we were not directly comparing responses between individuals or animals. Rather, we compared two methods for evaluation and no statistical conclusions were drawn. |
| Blinding | We were blinded to identifying information for all donor samples and therefore this information did not inform us in any way in choosing donor samples. |

# Reporting for specific materials, systems and methods

We require information from authors about some types of materials, experimental systems and methods used in many studies. Here, indicate whether each material, system or method listed is relevant to your study. If you are not sure if a list item applies to your research, read the appropriate section before selecting a response.

## Materials & experimental systems

| n/a | Involved in the study |
|---|---|
| ☐ | ☒ Antibodies |
| ☐ | ☒ Eukaryotic cell lines |
| ☒ | ☐ Palaeontology and archaeology |
| ☐ | ☒ Animals and other organisms |
| ☒ | ☐ Clinical data |
| ☒ | ☐ Dual use research of concern |
| ☒ | ☐ Plants |

## Methods

| n/a | Involved in the study |
|---|---|
| ☒ | ☐ ChIP-seq |
| ☒ | ☐ Flow cytometry |
| ☒ | ☐ MRI-based neuroimaging |

# Antibodies

| | |
|---|---|
| Antibodies used | Antibodies were isolated and purified from human and animal sera samples. mAbs CC6.30.2 and TXG-0078 were recombinantly expressed and purified as previously described and properly referenced. AP-conjugated AffiniPure goat anti-human IgG was used for ELISA experiments and was purchased from Jackson Immunoresearch (Cat # 109-055-097, Lot # 141947). |
| Validation | No novel antibodies were discovered or validated. |

# Eukaryotic cell lines

Policy information about cell lines and Sex and Gender in Research

| | |
|---|---|
| Cell line source(s) | FreeStyle293F (ThermoFisher Sci, Cat # A14528) |
| Authentication | No authentication. |
| Mycoplasma contamination | Mycoplasma is tested on a monthly basis. All cell lines used are confirmed negative. |
| Commonly misidentified lines (See ICLAC register) | No commonly misidentified cell lines were used in the study. |

# Animals and other research organisms

Policy information about studies involving animals; ARRIVE guidelines recommended for reporting animal research, and Sex and Gender in Research

| | |
|---|---|
| Laboratory animals | Healthy adult male or female C57BL/6J (CD45.2+/+) mice heterozygous for the BG18gH KI, generated as previously reported (39), and 8- to 12- week-old male B6.SJL-Ptprca Pepcb/BoyJ mice (CD45.1+/+) purchased from Jackson Laboratory were used in this study. Mice were housed at the animal facility, with free access to food and water, controlled temperature, and a 12:12 hours light-dark cycle. Mice were not involved in previous procedures and were drug and test naïve. |
| Wild animals | No wild animals were used in the study. |
| Reporting on sex | Male and Female. |
| Field-collected samples | No field collected samples were used in the study. |
| Ethics oversight | Institutional Animal Care and Use Committee (IACUC) of Harvard University and Massachusetts General Hospital (MGH), an Association for Assessment and Accreditation of Laboratory Animal Care International (AAALAC)–accredited facility, under Animal Study Protocols 2016N000286 and 2016N000022. |

Note that full information on the approval of the study protocol must also be provided in the manuscript.

# Plants

| | |
|---|---|
| Seed stocks | n/a |
| Novel plant genotypes | n/a |
| Authentication | n/a |

