## [Peer Review File · Nature Biomedical Engineering]

Microfluidics combined with electron microscopy for rapid and high-throughput mapping of antibody-viral glycoprotein complexes

Corresponding Author: Dr Andrew Ward

Version 0:

Decision Letter:

Dear Andrew,

Thank you again for submitting to *Nature Biomedical Engineering* your manuscript, "Rapid and High-Throughput Imaging of Immune Complexes Using Microfluidics". As noted in previous correspondence, the manuscript has been seen by four experts, yet one of them was unable to deliver any feedback. We then recruited an additional reviewer with substantial expertise in structural biology. The two reports I had already forwarded to you and the additional report are included at the end of this message.

You will see that the reviewers appreciate the work, and that they offer a few suggestions that I am hoping you will consider. In particular, please ensure that the methodology is thoroughly reported, to facilitate the reproducibility of the experimentation and findings.

When you are ready to resubmit your manuscript, please upload the revised files, a point-by-point rebuttal to the comments from all reviewers, the [reporting summary](https://www.nature.com/authors/policies/ReportingSummary.pdf), and a cover letter that explains the main improvements included in the revision and responds to any points highlighted in this decision.

Please follow the following recommendations:

- * Clearly highlight any amendments to the text and figures to help the reviewers and editors find and understand the changes (yet keep in mind that excessive marking can hinder readability).
- * If you and your co-authors disagree with a criticism, provide the arguments to the reviewer (optionally, indicate the relevant points in the cover letter).
- * If a criticism or suggestion is not addressed, please indicate so in the rebuttal to the reviewer comments and explain the reason(s).
- * Consider including responses to any criticisms raised by more than one reviewer at the beginning of the rebuttal, in a section addressed to all reviewers.
- * The rebuttal should include the reviewer comments in point-by-point format (please note that we provide all reviewers will the reports as they appear at the end of this message).
- * Provide the rebuttal to the reviewer comments and the cover letter as separate files.

We hope that you will find the referee reports helpful when revising the work, which we look forward to receive. Please do not hesitate to contact me should you have any questions.

Best wishes,

Pep

Pep Pàmies

Reviewer #2 (Report for the authors (Required)):

This manuscript describes ground-breaking new technology that will revolutionize the way we perform serum analyses in disease and vaccine research. A few years ago this research group developed the EMPEM method (polyclonal epitope mapping by EM) which was a major breakthrough in polyclonal serum antibody analysis as it visualizes different specificities within a polyclonal serum against a given antigen, highlighting immunodominant epitopes, changes in specificity over time etc. However, the technique is somewhat limited because it is quite time-consuming and sample-consuming. From experience I know that this limits the use of the technique for use in small animals (mice) and human samples as only small sera volumes are often available, in particular in historic bio-banked sample collections. The new microfluidics-based method negates these limitations, now allowing high throughput screening of serum antibody – antigen complexes. The manuscript itself very well written and clear.

I have no major concerns.

Minor comments.

1. The consensus abbreviation for Spike glycoprotein is S not S-GP

2. It would be useful to include a paragraph on limitations of the technique. Furthermore, what are the remaining bottlenecks in high throughput use of this technique? Number of channels on the device? Microscope access time? Computational power? Other?

Reviewer #3 (Report for the authors (Required)):

I must preface my review by saying that my background is NOT in structural biology. Hence I cannot comment on the quality of the EM-based resolution of epitopes. My lab does however perform a lot of epitope mapping studies. In that context, I emphatically agree with the authors that any improvement in throughput and reduction of laboriousness would be highly welcome. This is particularly true for polyclonal responses which are hard to deconvolute even when mutant antigens are available for mapping.

To my (perhaps somewhat untutored) eyes, the novelty of the authors' invention lies primarily in the use of a microfluidics device and in the optimization of the immunocapture/elution protocol, whereas everything that happens downstream of it is rather standard (though undoubtedly sophisticated). The recovery of particles appears to be similar for the microfluidics-based and the traditional method.

The more difficult question for me to answer relates to assessing the novelty of the authors' approach. My sentiment is that their method offers clear advantages in terms of reduced labor. Whether the costs and the throughput are massively reduced, is less clear to me. Certainly, the downstream analysis is still not trivial. The authors emphasize multiple times that their method enables high-throughput mapping (starting with the title: "Rapid and High-Throughput Imaging of Immune Complexes"). But what does "high throughput" exactly mean here? Their proof of high throughput relies on just four individual patients. Why is that? If the method really enables mapping within 90 minutes, it should have been possible to map hundreds (or even thousands) of patients - and make some truly significant discoveries in the process. The authors express the expectation that their method could be used for immediate epitope mapping of severely ill patients. That is a very ambitious goal, and the kind of throughput that would be required does not appear to have been proven by this study.

The mouse study is interesting and entails more datapoints, but in fairness one has to say that immunized mice typically sport enormous titers compared to what one can ever hope to see in humans, and therefore extrapolations are justified only up to a certain point.

My assessment is that the method appears to be robust, and the study is conducted in a very careful and professional manner. The authors should be commended for that. The magnitude of the advance over the state of the art is a bit more difficult for me to judge, and I shall defer to the Editor for a verdict.

Reviewer #4 (Report for the authors (Required)):

Sewall, Ward, and colleagues present a very exciting advance on their pioneering EMPEM work. They introduce a new microfluidics-based sample workup to more efficiently prepare immune complexes for single particle EM analyses. Two of the biggest advances from this are eliminating the need for Fab digestion and scaling back the required amount of input serum by a factor of 50-100. If the mEM setup/device could be easily implemented by others in the field, it could surely impact the serology field at large.

That being said, it is not always easy to assess how robust the identified Fab densities are in the reconstructed immune complexes, and therefore if their claim of improved sensitivity (vs EMPEM) towards a broader array of epitopes is true. Many of the displayed densities in the composite 3D maps have shapes that appear to deviate from the expected Fab structure. Reproducibility between repeated sample injections is not demonstrated, which could help to prove the robustness of the reconstructions. Rigid-body fits of antigen+fab in the displayed maps could help evaluate the quality of the putative Fab densities.

While the nsEM data presented seems most convincing, it is not clear if the mEM setup is as compatible with cryoEM analysis as claimed in the manuscript. Especially the limited resolution of the SARS-CoV-2 spike reconstruction is somewhat concerning in this regard. It is typically a very tractable target for which high resolution should be easily obtained. The manuscript might improve from a more transparent and fair discussion of the current limitations (or otherwise a more convincing demonstration of the suitability of mEM towards cryoEM).

Most importantly, I do not believe it is possible to exactly reproduce their device/setup with the materials provided in the current manuscript. For this development to truly advance the field, the manuscript should include a more detailed description of the design so that it can be adopted widely. Addition of technical drawings of the PDMS cell, PMMA clamp, and their assembly including tubing, sample pump, valves and collectors would be of great help.

While the manuscript presents a very exciting advance, the above points should be addressed in a revision before publication in my opinion.

Minor points:

- The manuscript repeatedly claims that the mEM analysis can be done in under 90 minutes, though it is unclear if this refers to just the microfluidics sample workup, or also includes the EM sample prep, data collection and analysis. If the latter is not the case, the claim of <90 minutes may be somewhat misleading

- Figure S4 appears to have several panels/labels mixed up.

Version 1:

Decision Letter:

Dear Andrew,

Thank you for your revised manuscript, "Rapid and High-Throughput Imaging of Immune Complexes Using Microfluidics". Having consulted with the original reviewers (whose comments you will find at the end of this message), I am pleased to write that we shall be happy to publish the manuscript in *Nature Biomedical Engineering*.

We will be performing detailed checks on your manuscript, and in due course will send you a checklist detailing our editorial and formatting requirements. You will need to follow these instructions before you upload the final manuscript files.

Best wishes,

Pep

Pep Pàmies

Chief Editor, Nature Biomedical Engineering

Reviewer #2 (Report for the authors (Required)):

The authors have addressed my comments. I have no further comments.

Reviewer #3 (Report for the authors (Required)):

I have read the new version of the manuscript and I have carefully analyzed the responses of the authors to my critique. The authors have done their best to address my points (and many other ones), but the facts are what they are. The throughput continues to be too small to make an impact on acute clinical bedside diagnostics. The improvement in the throughput over the previous iteration of the technology is significant but not revolutionary. Finally, the dilution of mouse sear 10 fold over

human sera does not reflect from the actual differences in titers for those autoantigens that can be expected in human patients. For all these reasons, I am a bit of the fence as to recommending the publication. Technically there is nothing wrong here but the technological advance is somewhat limited. I am aware that (1) this can be said of every single technology paper ever published and (2) I could be wrong. Hence I would rather defer this judgment call to the editor.

Reviewer #4 (Report for the authors (Required)):

All my original concerns have been addressed in this revised version. I would like to congratulate the authors on an outstanding piece of work.

Version 2:

Decision Letter:

Dear Dr Ward,

I am happy to inform you that your manuscript, "Microfluidics combined with electron microscopy for rapid and high-throughput mapping of antibody-viral glycoprotein complexes", has now been accepted for publication in *Nature Biomedical Engineering*.

Over the next few weeks, the figures will be checked for production quality, the text edited to ensure that it conforms to house style, and the manuscript typeset.

Our Articles are published about 40 days after the acceptance date (we recommend that you inform your institutional press office of this timeframe), and you will be notified of the actual publication date a few days in advance. Articles can be published any working day of the week, and are pushed live shortly after 10 am London time.

Publishing agreement. You will be asked to digitally sign a publishing agreement (grant of rights). After the signed publishing agreement has been received, the proofs of the article will be sent to you for review. If you have any queries during the production process, or you cannot meet the requested deadline for returning the proofs, please contact rjsproduction@springernature.com.

Nature Biomedical Engineering is a Transformative Journal. Authors may publish their research with us through the traditional subscription access route, or make their paper immediately open access through payment of an article-processing charge. More [information about publication options](https://www.springernature.com/gp/open-research/transformative-journals) is available.

You may need to take specific actions to [comply](https://www.springernature.com/gp/open-research/funding/policy-compliance-faqs) with funder and institutional open-access mandates. If the work described in the accepted manuscript is supported by a funder that requires immediate open access (as outlined, for example, by [Plan S](https://www.springernature.com/gp/open-research/plan-s-compliance)) and your manuscript was originally submitted on or after January 1st 2021, then you should select the gold OA route. Authors selecting subscription publication will need to accept our standard licensing terms (including our [self-archiving policies](https://www.springernature.com/gp/open-research/policies/journal-policies)), and these will supersede any other terms that the author or any third party may assert apply to any version of the manuscript.

Acceptance of your manuscript is conditional on agreement, by all authors, with both our [media embargo](http://www.nature.com/authors/policies/embargo.html) and [confidentiality and pre-publicity](http://www.nature.com/authors/policies/confidentiality.html) policies. In particular, you may arrange your own publicity of the Article (for instance, through your institutional press office), as long as you ensure that journalists strictly adhere to the media embargo.

To assist you in disseminating the work, as soon as the Article is published you will be able to take advantage of the Springer Nature [SharedIt](https://www.springernature.com/gp/researchers/sharedit) initiative to [generate a unique shareable link to the Article](http://authors.springernature.com/share) that will allow anyone (with or without a subscription) to read it. Recipients of the link who are subscribers will also be able to download and print the PDF.

Thank you for having submitted this work to *Nature Biomedical Engineering*.

Best wishes,

Barbara Cheifet
Editor
Nature Biomedical Engineering

Dear Dr. Pep Pàmies,

We hereby submit a revised version of our manuscript describing the development of a microfluidic system for the rapid and high-throughput imaging of immune complexes. We appreciate the reviewers' perspective on the significance of our study and your interest in considering our manuscript for publication. As outlined below, we have addressed the reviewers' concerns and provided more comprehensive information on the development of the microfluidic system, including the present limitations and a more thorough validation to ensure reproducibility of the method. As noted, detailed schematics comprising flow cell dimensions and system components are reported in an additional figure as well as described in the manuscript text. We hope that you share our enthusiasm for publication of this work.

Response to Reviewers for Manuscript nBME-24-1555-T

(Original reviewers' comments in black, our response in blue)

Reviewer #2 (Report for the authors):

This manuscript describes ground-breaking new technology that will revolutionize the way we perform serum analyses in disease and vaccine research. A few years ago this research group developed the EMPEM method (polyclonal epitope mapping by EM) which was a major breakthrough in polyclonal serum antibody analysis as it visualizes different specificities within a polyclonal serum against a given antigen, highlighting immunodominant epitopes, changes in specificity over time etc.

However, the technique is somewhat limited because it is quite time-consuming and sample-consuming. From experience I know that this limits the use of the technique for use in small animals (mice) and human samples as only small sera volumes are often available, in particular in historic bio-banked sample collections. The new microfluidics-based method negates these limitations, now allowing high throughput screening of serum antibody – antigen complexes. The manuscript itself very well written and clear.

I have no major concerns.

We sincerely appreciate the positive feedback and enthusiasm about the manuscript.

Minor comments.

1. The consensus abbreviation for Spike glycoprotein is S not S-GP

We thank the reviewer; this has now been fixed.

2. It would be useful to include a paragraph on limitations of the technique. Furthermore, what are the remaining bottlenecks in high throughput use of this technique? Number of channels on the device? Microscope access time? Computational power? Other?

We thank the reviewer for this suggestion, the revised manuscript includes a paragraph in the discussion on limitations of mEM including particle density and heterogeneity within the datasets which can affect resolution (page 8, line 337-343).

There are two main bottlenecks regarding throughput with the mEM technique. The first is the particle density of the immune complexes on the EM grid, which is now included in the Discussion section on limitations (page 8, lines 337-343). For ns-EM, this can be addressed by using a double blot technique when staining the sample on an EM grid which allows for an increase in the number particles being deposited (Methods section, page 24, lines 849-851). Double blotting together with collecting a larger dataset alleviates this bottleneck (particle density) and ensures enough particles go into the final 3D reconstructions. However, the lower particle numbers for these immune complexes make high-resolution cryo-EM of immune complexes challenging. Second, the number of channels within the flow cell limits the throughput of the mEM device. Currently, the flow cell contains a single channel that can be regenerated and reused repeatedly, which enables a single sample to be prepared in 90 minutes. An area of ongoing optimization involves increasing the number of channels within the flow cell to allow for multiple runs at once, which will ensure a significantly greater number of samples, up to 4 samples in parallel, can be processed within 90 minutes. We have included this information in the Discussion section (page 7, lines 333-334).

Reviewer #3 (Report for the authors):

I must preface my review by saying that my background is NOT in structural biology. Hence I cannot comment on the quality of the EM-based resolution of epitopes. My lab does however perform a lot of epitope mapping studies. In that context, I emphatically agree with the authors that any improvement in throughput and reduction of laboriousness would be highly welcome. This is particularly true for polyclonal responses which are hard to deconvolute even when mutant antigens are available for mapping.

To my (perhaps somewhat untutored) eyes, the novelty of the authors' invention lies primarily in the use of a microfluidics device and in the optimization of the immunocapture/elution protocol, whereas everything that happens downstream of it is rather standard (though undoubtedly sophisticated). The recovery of particles appears to be similar for the microfluidics-based and the traditional method.

The more difficult question for me to answer relates to assessing the novelty of the authors' approach. My sentiment is that their method offers clear advantages in terms of reduced labor. Whether the costs and the throughput are massively reduced, is less clear to me. Certainly, the downstream analysis is still not trivial. The authors emphasize multiple times that their method enables high-throughput mapping (starting with the title: "Rapid and High-Throughput Imaging of Immune Complexes"). But what does "high throughput" exactly mean here? Their proof of high throughput relies on just four individual patients. Why is that? If the method really enables mapping within 90 minutes, it should have been possible to map hundreds (or even thousands) of patients - and make some truly significant discoveries in the process.

The reviewer raises an important point about what constitutes high-throughput in this context. In the manuscript we state that mEM enables high-throughput structural assessment of the polyclonal antibody response to viral glycoproteins, which is in comparison to the number of samples that could be processed using the previous EMPEM method. With conventional EMPEM, a single sample requires one week of processing to go from sera to a purified immune complex ready for imaging by EM, however, a trained technician can process 5-6 samples at a time. Using mEM, four immune complexes can be readily prepared within a day and therefore within a week this method allows for 20 complexes to be prepared for imaging. Here, we chose to illustrate the throughput difference between mEM and conventional EMPEM using 16 samples, which by conventional EMPEM processing would take >3 weeks versus four days using mEM. In this study the 16 samples that were available included sera from four unique donors complexed with four different glycoprotein antigens, but it could have been sera from 16 unique donors complexed with one antigen and the processing time would have been identical.

With the current flow cell design and manual injections using a syringe pump, only a single sample can be run at a time using mEM. However, this is an area of ongoing optimization to produce a flow cell with additional channels such that the 90 minutes of sample preparation time can include running parallel samples in a single run, which will allow us to further increase the throughput of mEM up to the number of channels of the platform.

To address the reviewer's comment, we have now included additional description of the term "high-throughput" in the Results (page 5, line 220-222) and Discussion sections (page 7, line 325-332).

The authors express the expectation that their method could be used for immediate epitope mapping of severely ill patients. That is a very ambitious goal, and the kind of throughput that would be required does not appear to have been proven by this study.

The reviewer raises a valid point, although the throughput of mEM increases our ability to perform structural epitope mapping almost in real time (from sera to 3D reconstruction of immune complex in 1 day) and far above the previous EMPEM method, due to the lack of a high number of samples of severely ill patients we decided to be conservative with our discussion of these data (page 7, lines 317-321).

The mouse study is interesting and entails more datapoints, but in fairness one has to say that immunized mice typically sport enormous titers compared to what one can ever hope to see in humans, and therefore extrapolations are justified only up to a certain point.

We thank the reviewer for pointing this out. Indeed, the antibody responses of immunized mice tend to be much higher than in humans. In the manuscript we diluted the mice sera samples 100x compared to 10x dilution when using human sera in HCoV vaccinees. The use of mice samples was to prove that we can longitudinally assess epitopic landscapes

using small volumes of sera. We have clarified in the Results and Discussion sections that mEM was used to process samples with very minute volumes of sera, such as with individual mice, and antibody titer data was used only as a correlative at the individual mouse level (page 7, lines 298-300 and lines 325-332).

My assessment is that the method appears to be robust, and the study is conducted in a very careful and professional manner. The authors should be commended for that. The magnitude of the advance over the state of the art is a bit more difficult for me to judge, and I shall defer to the Editor for a verdict.

We sincerely appreciate the positive feedback from the reviewer on the manuscript.

Reviewer #4 (Report for the authors):

Sewall, Ward, and colleagues present a very exciting advance on their pioneering EMPEM work. They introduce a new microfluidics-based sample workup to more efficiently prepare immune complexes for single particle EM analyses. Two of the biggest advances from this are eliminating the need for Fab digestion and scaling back the required amount of input serum by a factor of 50-100. If the mEM setup/device could be easily implemented by others in the field, it could surely impact the serology field at large.

That being said, it is not always easy to assess how robust the identified Fab densities are in the reconstructed immune complexes, and therefore if their claim of improved sensitivity (vs EMPEM) towards a broader array of epitopes is true. Many of the displayed densities in the composite 3D maps have shapes that appear to deviate from the expected Fab structure. Reproducibility between repeated sample injections is not demonstrated, which could help to prove the robustness of the reconstructions. Rigid-body fits of antigen+fab in the displayed maps could help evaluate the quality of the putative Fab densities.

The reviewer raises a valid point that the low-resolution immune complexes have Fab densities that can be more difficult to interpret, especially considering that IgG is being complexed rather than purified Fab. To first answer the question of reproducibility, we used mEM to complex sera from Donor 74 with SARS-2 S two additional times and assessed epitope mapping in these duplicate complexes in comparison to the original complex in the paper. We observed a similar epitopic landscape across all three complexes with polyclonal antibodies identified targeting the three NTD sites, the RBD, SD2 and S2 regions, and added these results in Figure S8 and a sentence in the Results section (page 5, lines 208-211).

Next, to examine the Fab density fits we did rigid-body fits of the six glycoproteins (SARS-2 S, HKU1 S, OC43 S, HA H1, HA Influenza B and HIV Env N332-GT5) with a subset of polyclonal antibodies representing the determined epitopes for each glycoprotein. Except for HA Influenza B, structures have been previously determined for each of the glycoproteins and their subsequent PDB models were docked into the low pass filtered trimer densities. For all polyclonal epitopes determined for each glycoprotein, a human

polyclonal Fab model with a polyalanine backbone was docked into the Fab densities and relative fit was examined. We observed that the antigen and Fab models fit well in the determined densities despite the flexibility and heterogeneity of the polyclonal IgG. We added these results in Figures S13 & S14 and to the Results section (page 5, lines 211-213; page 6, lines 241-244 and 262-263; page 7, lines 294-296).

While the nsEM data presented seems most convincing, it is not clear if the mEM setup is as compatible with cryoEM analysis as claimed in the manuscript. Especially the limited resolution of the SARS-CoV-2 spike reconstruction is somewhat concerning in this regard. It is typically a very tractable target for which high resolution should be easily obtained. The manuscript might improve from a more transparent and fair discussion of the current limitations (or otherwise a more convincing demonstration of the suitability of mEM towards cryoEM).

Indeed, the reviewer raises an important point on the limited resolution of the SARS-CoV-2 spike obtained in the manuscript. Although the eluted protein concentration from mEM is readily compatible with ns-EM, cryo-EM often requires a concentration that is at least 10x greater. For all cryo-EM structures reported here we used graphene oxide grids to minimize the very high concentration needed for freezing. While this allowed high-resolution structures of apo HA and OC43, since SARS-CoV-2 spike fluctuates between two distinct conformations (all RBDs down versus one RBD up), increasing the heterogeneity within the dataset, high particle number would be needed to get high-resolution structures of the spike in the future. Further optimization of the capture surface in the mEM flow cell is time consuming but currently ongoing to improve the overall protein density before freezing.

The revised version of the manuscript includes a more detailed discussion on the described limitations of the mEM technique in both the Results (page 4, lines 164-165) and Discussion sections (page 8, lines 337-343).

Most importantly, I do not believe it is possible to exactly reproduce their device/setup with the materials provided in the current manuscript. For this development to truly advance the field, the manuscript should include a more detailed description of the design so that it can be adopted widely. Addition of technical drawings of the PDMS cell, PMMA clamp, and their assembly including tubing, sample pump, valves and collectors would be of great help.

We thank the reviewer for raising this important point and have now included detailed drawings and design schematics of the PDMS flow cell, PMMA clamp and the full mEM assembly in Figure S1.

While the manuscript presents a very exciting advance, the above points should be addressed in a revision before publication in my opinion.

We thank the reviewer for the positive feedback and informative insights. The revised manuscript includes additional supplemental data to address the issues raised by the reviewer. We detail these new data in the responses above.

Minor points:

- The manuscript repeatedly claims that the mEM analysis can be done in under 90 minutes, though it is unclear if this refers to just the microfluidics sample workup, or also includes the EM sample prep, data collection and analysis. If the latter is not the case, the claim of <90 minutes may be somewhat misleading

We thank the reviewer for bringing up this point. Indeed, the reference to 90 minutes only includes sample preparation time not data collection and analysis. We have now clarified throughout the text that “time” for mEM refers specifically to sample preparation time. Further, we included the total time for a single sample, including imaging and data processing, which is 1-1.5 days, in the revised manuscript as well (page 3, lines 112-116).

- Figure S4 appears to have several panels/labels mixed up.

We thank the reviewer; this has now been fixed. With the additional of Figure S1, this has now been renumbered as Figure S5.